# Geometric Graph Neural Diffusion for Stable Molecular Dynamics Simulations

**Haokai Hong [1], Wanyu Lin [1, 2]\*, Chusong Zhang [1, 3], Kay Chen Tan [1]**
[1]Department of Data Science and Artificial Intelligence, [2]Department of Computing,
The Hong Kong Polytechnic University, Hong Kong SAR, China. [3] Zhejiang University.
`haokai.hong@connect.polyu.hk`, {`wan-yu.lin,kctan`}`@polyu.edu.hk`

## Abstract

Geometric graph neural networks (Geo-GNNs) have revolutionized molecular dynamics (MD) simulations by providing accurate and fast energy and force predictions. However, minor prediction errors could still destabilize MD trajectories in real MD simulations due to the limited coverage of molecular conformations in training datasets. Existing methods that focus on in-distribution predictions often fail to address extrapolation to unseen conformations, undermining the simulation stability. To tackle this, we propose Geometric Graph Neural Diffusion (GGND), a novel framework that can capture geometrically invariant topological features, thereby alleviating error accumulation and ensuring stable MD simulations. The core of our framework is that it iteratively refines atomic representations, enabling instantaneous information flow between arbitrary atomic pairs while maintaining equivariance. Our proposed GGND is a plug-and-play module that can seamlessly integrate with existing local equivariant message-passing frameworks, enhancing their predictive performance and simulation stability. We conducted sets of experiments on the 3BPA and SAMD23 benchmark datasets, which encompass diverse molecular conformations across varied temperatures. We also ran real MD simulations to evaluate the stability. GGND outperforms baseline models in both accuracy and stability under significant topological shifts, advancing stable molecular modeling for real-world applications.

## 1 Introduction

Molecular dynamics (MD) simulations rely on force fields to approximate the underlying potential energy surface and generate long-temporal trajectories of molecular systems. Geometric graph neural networks (Geo-GNNs) have transformed MD simulations by providing a computationally efficient alternative to quantum mechanical methods, while maintaining high accuracy in predicting energies and forces (Wang et al., 2024a; Batatia et al., 2022; Wang et al., 2024c). Existing Geo-GNN evaluations mainly focus on the accuracy of predicting forces and overlook the performance evaluation in real MD simulations, e.g., whether the real MD could reveal detailed physical mechanisms (Lane et al., 2011). Recent studies have shown that even small errors in predicting forces could lead to catastrophic failure in real long-time simulations (Fu et al., 2023). This is because throughout the long-temporal trajectory, it can exhibit molecular conformations that are out of the training distribution. More specifically, due to the lack of extrapolation capability, most Geo-GNNs cannot produce accurate force prediction for unseen conformation, introducing pathological behaviors, i.e., unphysical chemical bonding, in a real MD simulation. Such a phenomenon can be quantified via chemical bonding connectivity in a real MD simulation, and is termed *stability* (Fu et al., 2023).

To examine the influence of conformation shifts on current Geo-GNNs, we use the 3BPA dataset (Kovács et al., 2021), as the dataset contains molecular geometries sampled at 300 K, 600 K, and 1200 K. Each temperature setting naturally induces a distinct conformation domain. We quantify conformation variations using edge-frequency distributions of atom pairs and prove that discrepancies grow systematically with increasing temperature gaps (Figure 1 (a) and (b). We then trained the representative Geo-GNNs VisNet (Wang et al., 2024c) and SEGNO (Liu et al., 2024) at

---

*Corresponding author.

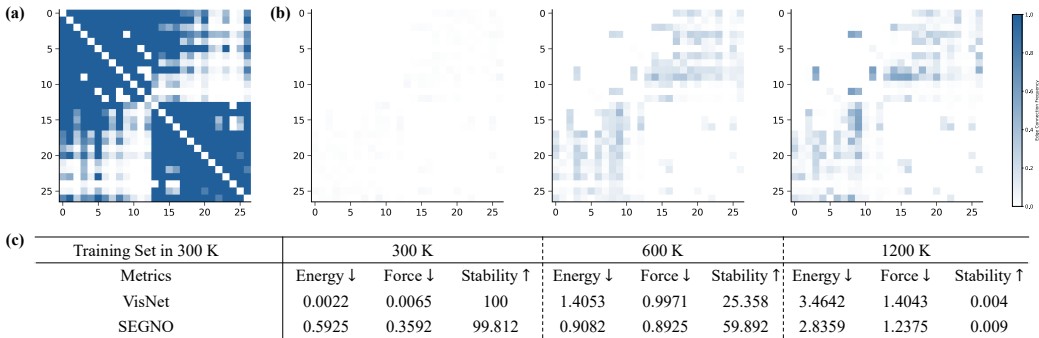

| Training Set in 300 K | 300 K | | | 600 K | | | 1200 K | | |
|---|---|---|---|---|---|---|---|---|---|
| Metrics | Energy ↓ | Force ↓ | Stability ↑ | Energy ↓ | Force ↓ | Stability ↑ | Energy ↓ | Force ↓ | Stability ↑ |
| VisNet | 0.0022 | 0.0065 | 100 | 1.4053 | 0.9971 | 25.358 | 3.4642 | 1.4043 | 0.004 |
| SEGNO | 0.5925 | 0.3592 | 99.812 | 0.9082 | 0.8925 | 59.892 | 2.8359 | 1.2375 | 0.009 |

Figure 1: **Geometric Topological Shift Analysis of the 3BPA Dataset and Extrapolation Performance Across Conformational Domains in the 3BPA Dataset**: (a) Distribution of adjacency matrix of 3BPA in training data (300 K); (b) distributional difference of Adjacency matrix of 3BPA in testing set ($k$=300 K, 600 K, and 1200 K) and training set (300 K); (c) extrapolation performance evaluation across conformational domains in the 3BPA Dataset.

300 K and tested them at 300 K, 600 K, and 1200 K. VisNet is the state-of-the-art Geo-GNN for simulating MD. While VisNet demonstrates strong within-domain performance (300 K), its accuracy degrades sharply under shifted conformation spaces (Figure 1 (c)). In contrast, SEGNO improves the generalizability via explicitly embedding physical biases. It indeed improves the extrapolation ability, but suffers from in-domain performance degradation. These findings confirm the urgent need for a new Geo-GNN that can remain robust and extrapolate effectively across various conformation domains, leading to stable MD simulations.

To fill the gap, we propose a new framework, dubbed geometric graph neural diffusion (GGND), inspired by the graph heat equation—a generalization of the diffusion equation rooted in spectral graph theory (Chung, 1997). Specifically, to facilitate the theoretical analysis, we first conceptualize domain variations in conformational spaces as "geometric topological shifts." Correspondingly, we introduce the diffusion process with two novel operators–equivariant gradient and diffusivity operators–to capture the invariance to conformational changes while maintaining equivariance. In particular, the gradient operator captures variations in node features across the topology of the geometric graph by characterizing differences between arbitrary nodes, while the diffusivity operator regulates the rate and extent of information propagation. Together, these operators drive the evolution of node representations, capturing all-pair information flows over a complete molecular graph, thereby remaining invariant to conformational changes. Our main contributions are outlined below:

*First*, we propose geometric graph neural diffusion (GGND) that can extrapolate effectively across various conformation domains, leading to stable MD simulations.

*Second*, we provide a theoretical analysis of GGND, establishing a regret bound under geometric topological shifts and proving the equivariance of the model. This regret-bound guarantees improved performance in extrapolating to unseen molecular conformations and enhances the stability of MD simulations.

*Third*, GGND functions as a plug-in module, seamlessly integrating with existing EGNNs to enhance their extrapolation capabilities. We evaluate GGND's performance on the 3BPA (Kovács et al., 2021) and SAMD23 (Kim et al., 2023) datasets, focusing on stability metrics for unseen molecular conformations. Our results demonstrate robust generalization across diverse conformational spaces and superior stability in real-world MD simulations compared to all baselines.

## 2 PRELIMINARIES AND RELATED WORKS

**Molecular graph.** In this paper, we explore the dynamics simulation of large-scale molecular systems, represented as a sequence of geometric graphs $\mathcal{G}$ indexed by time $t$. Suppose we have $N$ atoms in the system, then the molecular system $\mathcal{G}$ at each snapshot can be represented as a point cloud denoted as $\mathcal{G} = \langle X, H \rangle$, where $X = [\mathbf{x}_1; \ldots; \mathbf{x}_N] \in \mathbb{R}^{N \times 3}$ is the atom coordinate matrix

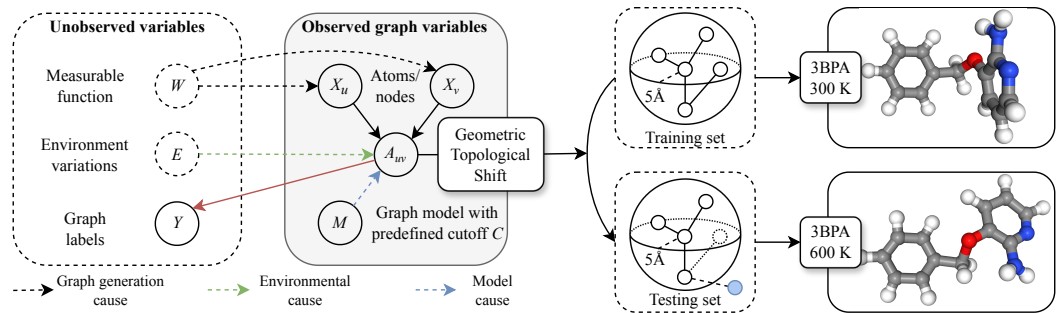

Figure 2: **The Illustration of Geometric Topological Shifts Caused by Environment $E$ and Model with Predefined Cutoff** $C$: The unobserved measurable function $W$ and environment $E$ (temperature or pressure), along with the modeling method $M$, serve as causes in the graph topology formation process, influencing unobserved variables and leading to variations in observed graph variables. This mechanism demonstrates how environmental causes (e.g., temperature changes) and model causes (predefined cutoff $C$) result in geometric topological shifts between the training set (e.g., 3BPA at 300 K) and the testing set (e.g., 3BPA at 600 K).

and $H = [\mathbf{h}_1; \dots; \mathbf{h}_N] \in \mathbb{R}^{N \times h}$ is the node feature matrix. $H$ typically contains atomic types or charge features, and it is generally time-invariant. Given the molecular structure $\mathcal{G}$, the objective of the machine learning force field is to predict the energy or forces with the molecular graph input $\mathcal{G}$.

**Topology of 3D graph.** In this study, we focus on the geometric topology of 3D molecular graphs. For geometric topology, nodes are atoms, and edges are established based on a predefined model-related radius cutoff distance threshold, such that pairs of atoms within this cutoff are considered neighbors. The term "topology" may also refer to the biochemical topology (or 2D molecular graph); however, unless specified otherwise, this paper focuses on geometric topology. Under conformational changes caused by the environment, the biochemical topology generally remains invariant, as it is defined by the fixed chemical connectivity of the molecule. Conversely, the geometric topology is dynamic, varying with the predetermined cutoff distance and the spatial coordinates of atoms, which may shift due to conformational changes.

**Geometric topological shift.** We propose a causal mechanism for geometric topology formation within a molecular system, as illustrated in Figure 2, building on prior work (Medvedev, 2014; Snijders & Nowicki, 1997). Unlike a 2D graph, our approach generalizes the data-forming mechanism to incorporate both geometric topological adjacency and node features. Specifically, a 3D graph with geometric topology, denoted as $\mathcal{G} = (X, H, A)$, is formed by a graphon—a continuous graph limit defined as a symmetric, measurable function $W : [0,1]^2 \to [0,1]$—serving as an unobserved latent variable, alongside a modeling method that specifies a cutoff radius $C$.

To elucidate the node-level structure of this graph, each node $u \in V$ is associated with an independent and identically distributed (i.i.d.) latent variable $U_u \sim \mathcal{U}[0,1]$. The vector and scalar features, $X = [X_u]$ and $H = [H_u]$, are random variables derived from each $U_u$ through node-wise functions $X_u = f(U_u; W)$ and $H_u = h(U_u; W)$, respectively. Next, the geometric topological adjacency matrix $A = [A(u,v)]$ is a random variable determined by a pairwise function $A(u,v) = h(U_u, U_v; W, E, M)$, which depends on the environment $E$ and the modeling method $M$. Changes in $E$, such as transitions from training to testing, lead to variations in the distribution of $A$. Beyond the node features and adjacency structure, the label $Y$ also varies due to conformational variations. We assume $Y$ is formed by a set function $Y = r(U_v \in V, A; W)$, with a specific realization denoted as $\mathbf{Y}$. We denote specific realizations of these random variables as matrices $\mathbf{X}$, $\mathbf{H}$, $\mathbf{A}$, and $\mathbf{Y}$.

**Extrapolation and stability.** Extrapolation remains a fundamental challenge in MD simulations, particularly in the application of data-driven machine learning methods to MD. From an ML perspective, extrapolation in MD can be categorized into two types: 1) extrapolation to chemical space, and 2) extrapolation to conformational space. The former entails predicting properties or dynamics for molecules absent from the training set, while the latter involves forecasting dynamics for un-

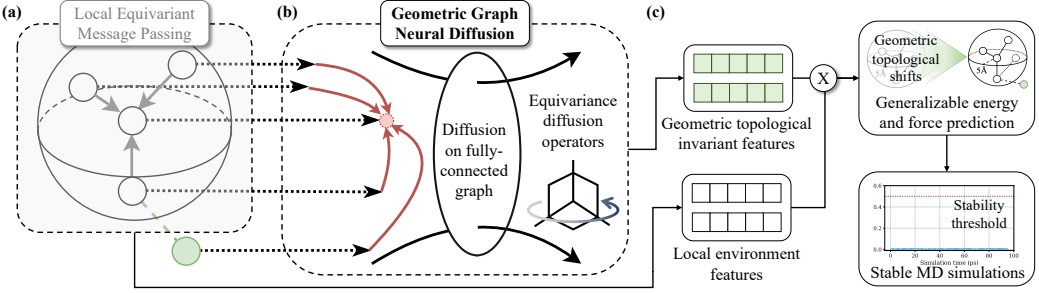

Figure 3: **The Illustration of Geometric Graph Neural Diffusion**: (a) Our method serves as a plug-in module that integrates with local equivariant message passing. (b) The GGND uses equivariant diffusion operators (gradient and diffusivity) on a fully connected graph to capture domain-invariant geometric topological features. (c) The local message passing and the equivariant diffusion operators are combined to address geometric topological shifts, enabling generalizable energy and force predictions for stable molecular dynamics simulations.

seen molecular geometries, such as samples from different temperatures or non-equilibrium states. Improving the ability to extrapolate to unseen conformations is critical for a model to ensure stable MD simulations, a principle supported by numerical MD methods (Barth & Schlick, 1998; Miao & Ortoleva, 2009). To tackle extrapolation, MatterSim (Yang et al., 2024) employs an active learning approach to address both categories, though it relies on costly high-quality data collection. Besides, SEGNO (Liu et al., 2024) integrates second-order motion laws to enhance the generalization of equivariant graph neural networks, yet it fails to address geometric topological shifts. Despite the importance of conformational space extrapolation for stable MD, enabling Geo-GNNs to generalize under geometric topological shifts remains an unresolved challenge. A brief overview of MD, generalization, and equivariance is provided in Appendix A.

## 3 METHOD

Our design enhances robustness to geometric and topological variations while preserving SE(3)-equivariance, thereby enabling stable MD simulations despite limited coverage of molecular conformations in training data. The model integrates two complementary components: (i) a novel geometric graph neural diffusion module with global attention and (ii) a conventional local equivariant message passing neural network (Satorras et al., 2021; Wang et al., 2024c). The GGND employs a diffusion process modeled as a partial differential equation (PDE) on the graph, incorporating global attention to capture long-range dependencies across all nodes. This global perspective mitigates challenges posed by geometric and topological shifts by facilitating information propagation beyond local neighborhoods. In contrast, the EGNN focuses on local interactions, updating node features and positions through message passing within local neighborhoods. The GGND serves as a plug-in module, seamlessly integrable with most existing EGNN frameworks to enhance their performance in stable MD simulations. We provide an overview of our method in Figure 3.

### 3.1 GEOMETRIC GRAPH NEURAL DIFFUSION

The geometric graph neural diffusion model is designed to learn equivariant features that are robust to shifts in geometric topology. To enable diffusion on geometric graphs, we incorporate higher-order equivariant message passing, which facilitates accurate modeling of such graphs. Given a geometric graph $\mathcal{G} = (\mathcal{V}, \mathcal{E})$ with $n = |\mathcal{V}|$ nodes, where each node $i \in \mathcal{V}$ has a scalar feature vector $\mathbf{h}_i \in \mathbb{R}^d$ and a position $\mathbf{x}_i \in \mathbb{R}^3$, and $\mathcal{E}$ is the set of edges determined by the adjacency matrix $\mathbf{A}_g$ assuming full connectivity in the graph. A concise background on diffusion processes on graphs is provided in Appendix B.

Each node $i$ has initial features $\mathbf{z}_i(0)$, which include chemical element features $\mathbf{h}_i$ (invariant scalars) and positions $\mathbf{x}_i$ (for equivariance). The features $\mathbf{z}_i(t)$ consist of spherical tensors labeled by irreducible representations of O(3), denoted as $\mathbf{z}_{i,kLM}(t)$, where $k$ indexes channels (learnable features), $L$ is the degree (e.g., $L = 0$ for invariants, $L = 1$ for vectors, and higher $L$ for ten-

sors), and $M = -L, \ldots, L$ indexes components. Diffusion models on geometric graphs replace discrete GNN layers with continuous time-evolving node embeddings $\mathbf{Z}(t) = \{\mathbf{z}_i(t)\}_{i=1}^n$, where $\mathbf{z}_i(t) : [0, \infty) \to \mathbb{R}^d$ and evolves according to the diffusion equation:

$$\frac{\partial \mathbf{Z}(t)}{\partial t} = \mathrm{div}\left[\mathbf{S}(\mathbf{Z}(t), \mathbf{X}, t) \odot \nabla \mathbf{Z}(t)\right], t \geq 0, \tag{1}$$

where $\mathbf{Z}(t) = \{\mathbf{z}_i(t)\}_{i=1}^n$ are equivariant node features, with initial conditions $\mathbf{Z}(0) = \phi_{\mathcal{E}}(\mathbf{X}, \mathbf{H})$, and $\phi_{\mathcal{E}}$ is the embedding layer through by radial basis functions (RBF).

The term $\mathbf{S}(\mathbf{Z}(t), \mathbf{X}, t)$ denotes the diffusivity over the graph, defined as an $n \times n$ matrix-valued function dependent on $\mathbf{A}_g$, which measures the rate of information flow between node pairs. The gradient $\nabla \mathbf{Z}$ maps node fields to edge fields, while the divergence operator $\mathrm{div}$ is its adjoint, mapping edge fields back to nodes. This diffusion process is modeled as a partial differential equation (PDE) on the graph, adapted to handle equivariant features via higher-order messages.

The GGND module is designed to learn features invariant to geometric topological shifts, enabling extrapolation to unseen molecular conformations. To achieve this, we introduce two novel operators: an **equivariant gradient operator** and an **equivariant diffusivity operator**, which facilitate global information flow while maintaining equivariance.

**Equivariant gradient operator.** The gradient operator $\nabla$ generalizes scalar differences to higher-order tensors, incorporating directional information to preserve SE(3)-equivariance. It is defined as:

$$(\nabla \mathbf{z})_{ij,kl_3m_3} = \sum_{\tilde{k}} W_{k\tilde{k}l_2}(\mathbf{z}_{j,\tilde{k}l_2m_2} - \mathbf{z}_{i,\tilde{k}l_2m_2}), \tag{2}$$

where $W$ are learnable weights for mixing channels. This equivariant gradient operator on the graph generalizes the scalar gradient to higher $L$, ensuring equivariance, with the difference $\mathbf{z}_j - \mathbf{z}_i$ modulated by directional information to preserve 3D structure. Notably, $j$ ranges over all nodes in $\mathcal{V}$, aligning with latent interactions among nodes determined by the underlying data manifold. This induces all-pair information flows over a complete graph and remains invariant to changes in $\mathcal{E}$ due to conformational variations.

**Equivariant diffusivity operator.** The diffusivity $\mathbf{S}(t)$ is made equivariant by defining it as a tensor-valued attention matrix. We extend scalar attention to tensors as follows:

$$\mathbf{S}(t)[i, j]_{kl_3m_3} = \sum_{l_1, l_2, m_1, m_2} C^{l_3m_3}_{l_1m_1,l_2m_2} R_{kl_1l_2l_3}(\|\mathbf{x}_{ji}\|) Y^{l_1}_{m_1}(\hat{\mathbf{x}}_{ji}) \phi(\mathbf{z}_i(t), \mathbf{z}_j(t))_{l_2m_2}, \tag{3}$$

where $C^{l_3m_3}_{l_1m_1,l_2m_2}$ are Clebsch-Gordan coefficients ensuring proper equivariance of $\mathbf{S}(t)$. Here, $\|\mathbf{x}_{ji}\| = \|\mathbf{x}_j - \mathbf{x}_i\|$, $\hat{\mathbf{x}}_{ji}$ is the unit vector, $Y^l_m$ are spherical harmonics (for directional equivariance), $R_{kl_1l_2l_3}$ is a learnable radial basis function derived from Bessel functions and an MLP (ensuring invariance to distance), and $\phi$ is an equivariant pairwise interaction (e.g., a gated tensor product). This formulation ensures that $\mathbf{S}(t)$ transforms correctly under SE(3), serving as an equivariant filter that captures global dependencies. The attention matrix $\mathbf{S} = (s(\mathbf{x}_i, \mathbf{x}_j))$ is right-stochastic, allowing Equation (1) to be rewritten as:

$$\frac{\partial \mathbf{Z}(t)}{\partial t} = (\mathbf{S}(\mathbf{Z}(t), \mathbf{X}, t) - \mathbf{I})\mathbf{Z}(t), \quad 0 \leq t \leq T. \tag{4}$$

Equation (4) governs the dynamics of the system from $t = 0$ to a specified stopping time $T$, producing geometric and topological node representations $\mathbf{Z}(T)$. The equation is generally nonlinear due to the dependence of the diffusivity matrix $\mathbf{S}$ on $\mathbf{Z}$. A linear variant emerges when attention weights are fixed; however, as static attention is impractical, we focus on the nonlinear GGND model.

**Output.** For energy prediction, we utilize the invariant components of $\mathbf{Z}(T)$, specifically $\mathbf{z}_{i,k00}(T)$, combined with local equivariant features learned by Equivariant Graph Neural Networks (EGNNs). This ensures that site-specific energy contributions $E_i$ remain invariant, computed as $E = \phi_{\mathcal{D}}(\mathbf{f}) = \sum_{i=1}^n \sum_{\tilde{k}} W_{\tilde{k}} \mathbf{f}_{i,\tilde{k}}$, where $\mathbf{f}_{i,\tilde{k}}$ represents features fused from local EGNN outputs $\mathbf{l}_{i,k}$ and geometric-topological invariant features $\mathbf{z}_{i,k00}(T)$ via concatenation and a linear transformation: $\mathbf{f}_{i,\tilde{k}} = W[\mathbf{l}_{i,k}; \mathbf{z}_{i,k00}(T)]$.

**Equivariance.** The diffusion process on the geometric graph, as described in Equation (4), enables the learning of domain-invariant features. With equivariant gradient and diffusivity operators, it also ensures the equivariance of the learned global features. We provide a proof of the equivariance of our GGND in Appendix C.

## 3.2 ALLEVIATING GEOMETRIC TOPOLOGICAL SHIFTS

We analyze the extrapolation capability of our geometric graph neural diffusion model with respect to geometric topological shifts, as defined in Section 2. Our focus is on the extrapolation error of the parametric function $\Gamma_\theta$, instantiated as the continuous equivariant diffusion model in Equations (4), when transferring from training data generated under environment $E_{\text{tr}}$ (and modeling method $M_{\text{tr}}$) to testing data under $E_{\text{te}}$ (and $M_{\text{tr}}$). Such shifts may arise from variations in adjacency matrices due to different cutoff radii or environmental conditions affecting inter-node distances in molecular systems.

Denote the training dataset of size $N_{\text{tr}}$ as $\{(\mathbf{X}^{(i)}, \mathbf{H}^{(i)}, \mathbf{A}^{(i)}, \mathbf{Y}^{(i)})\}_{i=1}^{N_{\text{tr}}}$, drawn from $p(X, H, A, Y \mid E = E_{\text{tr}}, M = M_{\text{tr}})$, and let $\ell(\cdot, \cdot)$ be a bounded loss function. The training error is

$$\mathcal{L}_{\text{tr}}(\Gamma_\theta; E_{\text{tr}}, M_{\text{tr}}) \triangleq \frac{1}{N_{\text{tr}}} \sum_{i=1}^{N_{\text{tr}}} \ell(\Gamma_\theta(\mathbf{X}^{(i)}, \mathbf{H}^{(i)}, \mathbf{A}^{(i)}), \mathbf{Y}^{(i)}). \tag{5}$$

Our objective is to minimize the expected loss on testing data from $p(X, H, A, Y \mid E = E_{\text{te}}, M = M_{\text{tr}})$:

$$\mathcal{L}(\Gamma_\theta; E_{\text{te}}, M_{\text{tr}}) \triangleq \mathbb{E}_{(\mathbf{X}', \mathbf{H}', \mathbf{A}', \mathbf{Y}') \sim p(X, H, A, Y \mid E = E_{\text{te}}, M = M_{\text{tr}})} [\ell(\Gamma_\theta(\mathbf{X}', \mathbf{H}', \mathbf{A}'), \mathbf{Y}')]. \tag{6}$$

When $E_{\text{te}} = E_{\text{tr}}$, this reduces to the standard in-distribution setting, where the extrapolation gap is bounded by

$$\mathcal{L}(\Gamma_\theta; E_{\text{tr}}, M_{\text{tr}}) - \mathcal{L}_{\text{tr}}(\Gamma_\theta; E_{\text{tr}}, M_{\text{tr}}) \leq \mathcal{D}_{\text{in}}(\Gamma_\theta, E_{\text{tr}}, M_{\text{tr}}, N_{\text{tr}}) = 2\mathcal{H}(\Gamma_\theta) + O\left(\sqrt{\frac{\log(1/\delta)}{N_{\text{tr}}}}\right). \tag{7}$$

With $\mathcal{H}(\Gamma_\theta)$, the Rademacher complexity of the function class is induced by $\Gamma_\theta$, and the upper bound is determined by dataset size and model complexity.

In the out-of-distribution regime where $E_{\text{te}} \neq E_{\text{tr}}$, geometric topological shifts complicate the analysis. Changes in geometric topologies alter node representations $\mathbf{Z}(T)$ in the equivariant graph diffusion equations (4), expressible as $\mathbf{Z}(T; \mathbf{A}) = f(\mathbf{Z}(0), \mathbf{A})$. The extrapolation gap could be decomposed into three terms (Wu et al., 2025). Assume $\ell$ and $\phi_\mathcal{D}$ are Lipschitz continuous. For geometric graph data generated per Section 2, with probability at least $1 - \delta$, the extrapolation gap satisfies:

$$|\mathcal{L}(\Gamma_\theta; E_{\text{te}}, M_{\text{tr}}) - \mathcal{L}_{\text{tr}}(\Gamma_\theta; E_{\text{tr}}, M_{\text{tr}})| \leq \mathcal{D}_{\text{in}}(\Gamma_\theta, E_{\text{tr}}, M_{\text{tr}}, N_{\text{tr}})$$
$$+ \mathcal{O}\left(\mathbb{E}_{\mathbf{A} \sim p(A \mid E_{\text{tr}}, M_{\text{tr}}), \mathbf{A}' \sim p(A \mid E_{\text{te}}, M_{\text{tr}})} [\|\mathbf{Z}(T; \mathbf{A}') - \mathbf{Z}(T; \mathbf{A})\|_2]\right) \tag{8}$$
$$+ \mathcal{O}\left(\mathbb{E}_{(\mathbf{A}, \mathbf{Y}) \sim p(A, Y \mid E_{\text{tr}}, M_{\text{tr}}), (\mathbf{A}', \mathbf{Y}') \sim p(A, Y \mid E_{\text{te}}, M_{\text{tr}})} [\|\mathbf{Y}' - \mathbf{Y}\|_2]\right).$$

We denote the first $\mathcal{O}(\cdot)$ as OOD model error $\mathcal{D}_{\text{M}}(\Gamma_\theta, E_{\text{tr}}, M_{\text{tr}}, E_{\text{te}})$ and the second $\mathcal{O}(\cdot)$ as OOD label error $\mathcal{D}_{\text{L}}(E_{\text{tr}}, M_{\text{tr}}, E_{\text{te}})$. Since $D_{\text{in}}$ is independent of testing data under $E_{\text{te}} \neq E_{\text{tr}}$, the impact of geometric topological shifts on extrapolation hinges on $\mathcal{D}_{\text{M}}$ and $\mathcal{D}_{\text{L}}$: the former captures variation in $\mathbf{Z}(T; \mathbf{A})$ due to shifting topologies (e.g., adjacency changes from varying cutoff radii or conformations), while the latter reflects label differences across environments or methods. $D_{\text{L}}$ is dictated by the data-generating process, whereas $\mathcal{D}_{\text{M}}$ depends on $\Gamma_\theta$, specifically the sensitivity of representations to shifts. We next examine $\Gamma_\theta$ as in Equation (4), adapted for equivariance.

**Theorem 3.1** *For geometric graph data per Section 2, if $f$ and $h$ are injective, the geometric graph neural diffusion model in Equation (4) reduces the representation variation $\|\mathbf{Z}(T; \mathbf{A}') - \mathbf{Z}(T; \mathbf{A})\|_2$ to any order $\mathcal{O}(\psi(\|\Delta\tilde{\mathbf{A}}\|_2))$, where $\psi$ is an arbitrary polynomial, $\Delta\tilde{\mathbf{A}} = \tilde{\mathbf{A}}' - \tilde{\mathbf{A}}$, and $\tilde{\mathbf{A}} = \mathbf{D}^{-1/2}\mathbf{A}\mathbf{D}^{-1/2}$ (with $\mathbf{A}$ incorporating geometric distances via cutoff radius and $\mathbf{D}$ is the diagonal degree matrix of $\mathbf{A}$).*

This indicates that the geometric graph neural diffusion model controls representation changes at arbitrary rates relative to $\|\Delta\tilde{\mathbf{A}}\|_2$, maintaining robust force prediction for conformation variations in molecular dynamics. The injectivity of $f$ and $h$ are mild assumptions, mapping from compact latent spaces to high-dimensional vector and scalar features. Applying Equation (8) yields the following.

**Corollary 3.2** *Under the condition of Theorem 3.1, the model-dependent extrapolation bound in Equation (8) reduces to arbitrary polynomial orders with respect to geometric topological shifts:*

$$\mathcal{D}_{\mathrm{M}}(\Gamma_\theta, E_{\mathrm{tr}}, E_{\mathrm{te}}, M_{\mathrm{tr}}) = O\left(\mathbb{E}_{\mathbf{A}\sim p(A|E_{\mathrm{tr}}, M_{\mathrm{tr}}), \mathbf{A}'\sim p(A|E_{\mathrm{te}}, M_{\mathrm{tr}})}[\psi(\|\Delta\tilde{\mathbf{A}}\|_2)]\right).$$

This bound ensures controllable extrapolation error at any rate relative to $\|\Delta\tilde{\mathbf{A}}\|_2$. The model achieves desired extrapolation ability under shifts, such as in machine learning force fields or simulations with conformational changes or varying cutoffs. In contrast, the change rate of features produced by the local message passing model has an exponential upper bound. We presented the proof for the Corollary 3.2 in the Appendix D.

## 4 EXPERIMENTS

### 4.1 EXPERIMENTAL SETUP

**Datasets.** We used the 3BPA and SAMD23 datasets to evaluate our model's performance, particularly in the presence of geometric and topological shifts. The 3BPA dataset consists of 500 training structures of the flexible, drug-like molecule 3-(benzyloxy)pyridin-2-amine at 300 K, with test data provided at 300 K, 600 K, 1200 K, and different dihedral angles (Kovács et al., 2021). The SAMD23 dataset comprises simulations of the semiconductor materials SiN and HfO under various conditions, including variations in initial structures, stoichiometry, temperature, strain, and defects, with unit cells containing up to 510 atoms (Kim et al., 2023).

**Baselines.** Our proposed equivariant graph neural diffusion can be integrated with any local equivariant message-passing-based method. To evaluate performance improvements, we selected four representative methods—NequIP (Batzner et al., 2022), MACE (Batatia et al., 2022), SEGNO (Liu et al., 2024), and VisNet (Wang et al., 2024c)—as baselines and compared our approach when combined with them against these baselines alone. Additionally, we conducted a comprehensive comparison with several SOTA models on the SAMD23 dataset, including Allegro (Musaelian et al., 2023), Equiformer V2 (Liao et al., 2024), QuinNet (Wang et al., 2023), Neural P$^3$M (Wang et al., 2024b), LSRM (Li et al., 2024b), and FreeCG (Shao et al., 2025),

**Metrics.** *Accuracy:* we evaluate the predictive performance of our model using the mean absolute error (MAE) for energy and force predictions. For the SAMD23 dataset, which includes SiN molecules with atom counts ranging from 16 to 510, we report the energy per atom to ensure comparability across molecular sizes. *Stability:* following the methodology in (Fu et al., 2023), we assess the stability of flexible molecules by monitoring bond length deviations. A real MD simulation is classified as unstable at time $T$ if the maximum deviation of any bond length from its equilibrium value exceeds a threshold, formally defined as: $\max_{i,j\in\mathcal{B}}|\|\mathbf{x}i(T) - \mathbf{x}j(T)| - b_{ij}| > \Delta$, where $\mathcal{B}$ denotes the set of all bonds, $i$ and $j$ are bond endpoints, $b_{ij}$ is the equilibrium bond length, and $\Delta$ is the stability threshold. For systems with periodic boundary conditions, stability is evaluated using the radial distribution function (RDF). A simulation is deemed unstable at time $T$ when: $\int_0^\infty \|\langle\mathrm{RDF}(r)\rangle - \langle\hat{\mathrm{RDF}}t(r)\rangle t = T^{T+\tau}\|dr > \Delta$, where $\langle\cdot\rangle$ represents the time-averaging operator, $\tau$ is a 1 ps time window, and $\Delta$ is set to 1.0. We perform constant-energy (NVE) molecular dynamics simulations at the specified temperature, employing Velocity Verlet integration over 100 ps with a 1 fs timestep. The stability metric is defined as the first timestep (in ps, ranging from 0 to 100) at which an unstable molecular configuration occurs. We conduct five independent molecular dynamics simulations and report the average stability metric as the final result. Higher stability values indicate better performance in maintaining long-term stable molecular dynamics simulations.

### 4.2 RESULTS AND ANALYSIS

Table 1: **Accuracy and Stability on the 3BPA Dataset.** MAE for energy (E, eV), force (F, eV/Å), and stability (S, ps) of three baseline models and our proposed model (+GGND), trained on configurations of the flexible drug-like molecule 3BPA at 300 K and evaluated on 300 K, 600 K, 1200 K, and varied dihedral angles. Best results are in **bold**; tied results are underlined.

| Conformation | Metrics | MACE | +GGND | NequIP | +GGND | SEGNO | +GGND | VisNet | +GGND |
|---|---|---|---|---|---|---|---|---|---|
| 300K | E (↓) | 0.113 | **0.010** | 0.165 | **0.094** | 0.593 | **0.293** | 0.002 | 0.002 |
| | F (↓) | 0.165 | **0.022** | 0.113 | **0.104** | 0.359 | **0.183** | 0.006 | 0.006 |
| | S (↑) | 100 | 100 | 100 | 100 | 99.812 | **100** | 100 | 100 |
| 600K | E (↓) | 0.161 | **0.023** | 0.335 | **0.122** | 0.908 | **0.295** | 1.405 | **0.022** |
| | F (↓) | 0.335 | **0.044** | 0.161 | **0.153** | 0.893 | **0.193** | 0.997 | **0.041** |
| | S (↑) | 100 | 100 | 98.271 | **100** | 59.892 | **100** | 25.358 | **100** |
| 1200K | E (↓) | 0.271 | **0.109** | 0.770 | **0.477** | 2.836 | **0.503** | 3.464 | **0.583** |
| | F (↓) | 0.770 | **0.111** | 0.271 | **0.269** | 1.238 | **0.285** | 1.404 | **0.304** |
| | S (↑) | 1.965 | **29.218** | 0.018 | **17.052** | 0.009 | **16.201** | 0.004 | **11.209** |
| Dihedral Slices | E (↓) | 0.169 | **0.012** | 0.387 | **0.375** | 0.923 | **0.267** | 0.789 | **0.050** |
| | F (↓) | 0.289 | **0.017** | 0.242 | **0.189** | 0.795 | **0.192** | 0.697 | **0.039** |
| | S (↑) | 100 | 100 | 89.119 | **100** | 72.282 | **100** | 47.785 | **100** |

Table 2: **Accuracy and Stability on the 3BPA Dataset.** MAE for energy per atom (E/A, eV), force (F, eV/Å), and stability (S, ps) obtained by SOTA models and our proposed model (GGND), trained on SiN and HfO semiconductor molecular system. Best results are in **bold**.

| Molecule | Splits | Metrics | NequIP | MACE | Allegro | Neural P3M | QuinNet | Equiformer V2 | LSRM | FreeCG | GGND |
|---|---|---|---|---|---|---|---|---|---|---|---|
| SiN | Test | E/A (↓) | 0.013 | 0.012 | 0.015 | 0.010 | 0.010 | 0.010 | 0.010 | 0.011 | **0.009** |
| | | F (↓) | 0.598 | 0.526 | 0.673 | 0.485 | 0.490 | 0.451 | 0.490 | 0.494 | **0.443** |
| | | S (↑) | 69.009 | 78.845 | 63.583 | 88.280 | 83.286 | 98.284 | 81.000 | 84.500 | **100** |
| | OOD | E/A (↓) | 0.022 | 0.018 | 0.028 | 0.016 | 0.017 | 0.021 | 0.018 | 0.018 | **0.015** |
| | | F (↓) | 1.018 | 0.912 | 1.185 | 0.837 | 0.836 | 0.972 | 0.832 | 0.844 | **0.754** |
| | | S (↑) | 63.733 | 65.710 | 55.824 | 85.888 | 86.512 | 82.031 | 74.217 | 76.631 | **99.892** |
| HfO | Test | E/A (↓) | 0.007 | 0.006 | 0.007 | 0.006 | 0.006 | **0.005** | 0.006 | 0.006 | **0.005** |
| | | F (↓) | 0.377 | 0.335 | 0.385 | 0.311 | 0.304 | 0.298 | 0.312 | 0.315 | **0.179** |
| | | S (↑) | 65.377 | 78.054 | 64.282 | 90.432 | 89.034 | 97.184 | 87.353 | 85.040 | **100** |
| | OOD | E/A (↓) | 0.011 | 0.010 | 0.012 | 0.009 | 0.009 | 0.010 | 0.010 | 0.009 | **0.008** |
| | | F (↓) | 0.430 | 0.570 | 0.593 | 0.459 | 0.457 | 0.683 | 0.544 | 0.593 | **0.279** |
| | | S (↑) | 61.621 | 65.689 | 60.982 | 84.209 | 85.453 | 79.762 | 86.373 | 75.916 | **97.928** |

**Performance on 3BPA datasets.** Experimental results on the 3BPA dataset in Table 1 show that integrating the proposed GGND module with baseline Geo-GNN models (MACE, NequIP, SEGNO, and VisNet) significantly enhances performance, particularly in extrapolating to conformational domains with improved stability. On the in-domain 300 K test set, GGND improves energy and force prediction accuracy for most baselines while achieving perfect stability at 100 ps. For instance, SEGNO's energy MAE decreases from 0.593 eV to 0.293 eV and force MAE from 0.359 eV/Å to 0.183 eV/Å, with stability rising from 99.812 ps to 100 ps. At 600 K, where domain shifts occur, GGND's advantages are more pronounced; it reduces VisNet's energy MAE from 1.405 eV to 0.022 eV and force MAE from 0.997 eV/Å to 0.041 eV/Å, boosting stability from 25.358 ps to 100 ps. Comparable improvements are observed for SEGNO, with energy MAE dropping from 0.908 eV to 0.295 eV and stability from 59.892 ps to 100 ps. These findings indicate that SEGNO addresses universal generalization but not geometric topological shifts. Additionally, GGND outperforms all baselines on dihedral slices.

Under severe geometric topological shifts at 1200 K, baselines suffer catastrophic degradation in stability (e.g., VisNet at 0.004 ps, MACE at 1.965ps, SEGNO at 0.009 ps), whereas GGND restores robustness, increasing MACE's stability to 29.218 ps (15-fold), NequIP to 17.052 ps (947-fold), SEGNO to 16.201 ps (1800-fold), and VisNet to 11.209 ps (2802-fold). Concurrent accuracy gains include VisNet's energy MAE reduction from 3.464 eV to 0.108 eV. These results highlight GGND's efficacy in mitigating geometric topological shifts via all-pair information diffusion, facilitating stable long-term MD simulations in unseen conformations without additional DFT data.

**Performance on SAMD23 dataset.** The GGND model outperforms baselines across SiN and HfO datasets in both Test and OOD splits, as shown in Table 2. For SiN, GGND achieves a lower en-

Table 3: **Ablation Analysis on the 3BPA Dataset.** MAE for energy (E, eV), force (F, eV/Å), and stability (S, ps) of baseline model, GGND, and two variants of GGND. Best results are in **bold**; tied results are underlined.

| Conformation | 300 K | | | 600 K | | | 1200 K | | |
|---|---|---|---|---|---|---|---|---|---|
| Variations | E ($\downarrow$) | F ($\downarrow$) | S ($\uparrow$) | E ($\downarrow$) | F ($\downarrow$) | S ($\uparrow$) | E ($\downarrow$) | F ($\downarrow$) | S ($\uparrow$) |
| Baseline | **0.002** | **0.006** | **100** | 1.405 | 0.997 | 25.358 | 3.464 | 1.404 | 0.004 |
| GGND † | 0.013 | 0.058 | **100** | 0.982 | 0.998 | 39.075 | 3.049 | 1.406 | 0.291 |
| GGND ‡ | 0.015 | 0.072 | 98.827 | 0.643 | 0.661 | 69.292 | 1.908 | 0.882 | 2.892 |
| GGND | **0.002** | **0.006** | **100** | **0.022** | **0.041** | **100** | **0.583** | **0.304** | **11.209** |

†: GGND with local diffusion on graph.
‡: Local message passing baseline plus fully-connected message passing.

ergy per atom (E/A) error of 0.009 eV and a force MAE of 0.443 eV/Å in the Test split, improving force predictions by approximately 9% over Neural P$^3$M and LSRM, with a perfect stability score of 100 ps. In the OOD split, GGND maintains robust performance with a stability score of 99.89 ps, significantly surpassing QuinNet and Neural P$^3$M. For HfO, GGND records an E/A of 0.005 eV and a force MAE of 0.179 eV/Å in the Test split, reducing force errors by over 40% compared to Neural P$^3$M and QuinNet, and achieving a perfect stability score of 100 ps. In the OOD split, its stability score of 97.93 ps notably exceeds baselines. GGND's equivariant diffusion process effectively captures all-pair interactions, ensuring insensitivity to conformational changes and enhancing stability in molecular dynamics simulations. The experimental results highlight the remarkable ability of GGND to address geometric topological shifts, as evidenced by its consistent outperformance across the SiN and HfO datasets. The model's outstanding stability scores of 100 ps in both Test splits and near-perfect scores in OOD splits (99.892 ps for SiN and 97.928 ps for HfO) suggest that the equivariant diffusion process effectively captures all-pair information flows, making GGND insensitive to conformational changes.

**Stability Visualization and Analysis.** The stability metric indicates the first time step at which the MD simulation becomes unstable. To better characterize the stability throughout the entire MD process, we visualize the maximum bond length deviation in Figure 4. In 100 ps MD simulations on the 3BPA dataset, GGND outperforms the ML-based baselines by maintaining stability. Notably, although GGND exhibits instability around 30 ps, these unstable states occur randomly, whereas both VisNet and MACE show persistent instability after a certain time step.

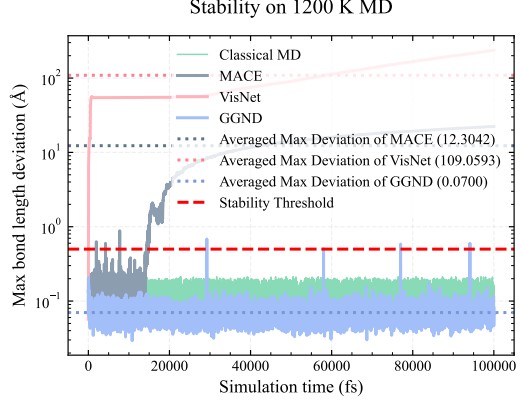

Figure 4: Stability of MD Simulations on 3BPA.

**Ablation Study.** To evaluate the impact of fully-connected diffusion in our proposed GGND model, we curated two variants: GGND†, which uses local diffusion on the graph, and GGND‡, which combines the baseline with fully-connected message passing. The ablation study on the 3BPA dataset (Table 3) demonstrates the superior generalization of GGND to unseen conformational domains at 600 K and 1200 K, while matching the baseline's optimal performance at 300 K (energy MAE: 0.002 eV, force MAE: 0.006 eV/Å, stability: 100 ps). In contrast, GGND†, limited by local diffusion, fails to generalize effectively, with performance close to the baseline (e.g., stability of 0.291 ps at 1200 K), as it cannot capture all-pair interactions. GGND‡ shows some generalization potential (e.g., stability of 2.892 ps at 1200 K) but underperforms GGND due to training challenges, highlighting the advantage of fully-connected diffusion in enabling robust, equivariant information flow for stable and accurate molecular dynamics simulations across diverse conformations.

## 5 CONCLUSION

In this study, we investigate the stability of MD simulations and identify extrapolation to unseen conformations as a key challenge. To address this, we propose GGND, a novel framework that improves the stability and generalizability of MD simulations by capturing geometrically invariant topological features through an equivariant diffusion process. By mitigating geometric topological shifts arising from conformational variations, GGND reduces error accumulation, ensures robust energy and force predictions for unseen molecular conformations, leading to stable molecular dynamics simulations. Our theoretical analysis establishes a regret bound under such shifts, providing formal guarantees of stability. Designed as a plug-and-play module, GGND integrates seamlessly with existing local equivariant message-passing networks, boosting out-of-domain performance while preserving in-domain accuracy. Comprehensive experiments on the 3BPA and SAMD23 datasets show that GGND surpasses baseline models in both accuracy and simulation stability.

### ACKNOWLEDGMENTS

This work was supported in part by the Hong Kong Research Grants Council General Research Fund Under Ref. No 15208725, the Hong Kong Polytechnic University Internal Research Fund Under P0057774, the Research Grants Council of the Hong Kong SAR (Grant No. PolyU15215623, PolyU15229824, C5052-23G, and SRFS2526-5S04), and the Hong Kong Polytechnic University (P0058445).

### ETHICS STATEMENT

This study adheres to the ICLR Code of Ethics, with careful consideration of the ethical implications of our work, particularly its societal impacts, which are comprehensively addressed in Appendix J. Our methodology does not involve human participants, sensitive data, or applications with significant misuse potential. We have prioritized fairness and transparency in the development of our models and findings, addressing potential biases in the dataset and model design in the referenced appendix. No conflicts of interest or funding concerns compromise the integrity of this research. The use of large language models is detailed in Appendix K.

### REPRODUCIBILITY STATEMENT

Complete proofs for all theoretical claims are included in Appendices D and C. This study utilizes the publicly available datasets 3BPA and SAMD23, accessible at https://pubs.acs.org/doi/10.1021/acs.jctc.1c00647 and https://github.com/SAITPublic/MLFF-Framework, respectively. We adhere to the data splits specified in the publications associated with these datasets, with all relevant parameters documented in Appendix E. The code is available at https://github.com/HaokaiHong/GGND. These resources collectively enable full replication of our experiments and results.

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

APPENDIX

## A RELATED WORKS

### A.1 MACHINE LEARNING MOLECULAR DYNAMICS SIMULATION AND ITS GENERALIZATION

Machine-learning for molecular dynamics simulation, including ML interatomic potentials and ML force fields, has emerged as an accurate and computationally efficient surrogate for quantum mechanical calculations in MD simulations and related atomistic modeling tasks. However, the generalization of ML methods for MD simulations remains unaddressed.

A major advance in improving the generalization of these models has been the development of "universal" or pretrained interatomic potentials trained on chemically diverse and large-scale datasets. Models such as M3GNet (Chen & Ong, 2022) and CHGNet (Deng et al., 2023) show that broad coverage of elements, bonding motifs, and structures can substantially improve transferability and reduce retraining costs. Nevertheless, these approaches often falter when faced with distribution shifts outside the training domain, such as conformations sampled under different thermodynamic conditions.

Another complementary line of work focuses on architectural inductive biases, especially equivariant geometric graph neural networks that explicitly encode physical symmetries. Equivariant message passing architectures, including recent efficient implementations (Batatia et al., 2022; Wang et al., 2024c), typically deliver higher accuracy and better sample efficiency than non-equivariant baselines. While these models interpolate well within the training regime, symmetry constraints alone are insufficient to guarantee robustness under changes in the conformational distribution.

Alongside architectural innovations, there is growing recognition that standard energy and force test errors can be misleading as proxies for MD stability. Benchmarking studies such as (Fu et al., 2023) emphasize trajectory-level evaluations, including long-term stability, conservation laws, and reproduction of thermodynamic observables. These evaluations often reveal substantial degradation under out-of-distribution (OOD) conditions even when pointwise prediction errors remain low.

To address such failures, some works integrate active learning and on-the-fly adaptation into MLFF/MLIP workflows. Active learning pipelines and uncertainty-aware simulation controllers selectively query new data in high-uncertainty regions (Yang et al., 2024), while recent methods such as TAIP (Cui et al., 2025) perform test-time adaptation to reduce the impact of train–test distribution gaps. These strategies mitigate specific failure modes but can be computationally demanding and lack systematic mechanisms to detect and react to distribution shifts in a theoretical manner.

Our work builds on these threads by proposing a geometric GNN that detects and responds to distribution shifts in conformational space, aiming to improve stability and accuracy when generalizing across thermodynamic regimes.

### A.2 GENERALIZABLE GRAPH NEURAL NETWORK

Recent advancements in graph neural network (GNN) generalization have progressed along four complementary dimensions: theoretical foundations, architectural innovations, training methodologies, and data-centric strategies. Theoretical studies have established sample-complexity and stability bounds, elucidating structural factors—such as propagation depth and graph connectivity—that influence generalization and inform design decisions (Tang & Liu, 2023; Yang et al., 2023). Architectural advancements demonstrate that incorporating inductive biases or attention mechanisms enhances generalization performance (Liu et al., 2024; Wu et al., 2025; Li et al., 2024a). Practical training and data-centric approaches translate theoretical insights into practice through causal learning and data augmentation techniques, effectively mitigating empirical generalization gaps (Abbahaddou et al., 2025; Fan et al., 2024). Collectively, these efforts outline a coherent research agenda: leverage theoretical insights to identify generalization bottlenecks, design architectures with inductive biases to reduce sample complexity, and employ training and data strategies, alongside out-of-distribution (OOD)-aware mechanisms, to ensure robust generalization.

However, existing research on GNN generalization has largely overlooked the challenge of conformational distribution drift in MLFF or MD. Specifically, environmental factors, such as temperature, can induce drifts in the geometric structure distribution of molecular graphs, which subsequently alter their topological configurations.

### A.3 EQUIVARIANT GRAPH NEURAL NETWORK

Physical systems obey symmetry principles that constrain valid model behavior. For molecular modeling, the governing laws are invariant or equivariant under the Euclidean group SE(3), including rotations, translations, and reflections. Neural networks that respect these symmetries have demonstrated substantial improvements in data efficiency, physical consistency, and generalization for atomistic modeling.

Equivariant graph neural networks enforce transformation consistency by ensuring that node representations transform according to group representations when input coordinates are transformed. These approaches can be divided into incorporating symmetry through invariant features derived from interatomic distances (Satorras et al., 2021; Xu et al., 2024) and propagating higher-order geometric tensors and directional information through equivariant message passing (Liao & Smidt, 2023; Batatia et al., 2022; Batzner et al., 2022). These architectures enable accurate modeling of energies and forces while preserving physically meaningful coordinate dependencies. Equivariant graph neural networks has been validated effective for a range of molecular modeling tasks, including force field prediction (Batatia et al., 2022; Batzner et al., 2022; Wang et al., 2024c), molecular dynamics simulation (Xu et al., 2024; Han et al., 2022), and molecular design (Hoogeboom et al., 2022; Hong et al., 2025a;b).

In molecular dynamics simulation, equivariance is particularly critical because force predictions must transform consistently with spatial rotations to ensure stable trajectory integration. Consequently, modern geometric deep learning approaches for interatomic potentials and force fields widely adopt equivariant representations as a fundamental inductive bias. However, symmetry preservation alone does not guarantee robustness under distributional shifts in geometric topology. Our work builds upon equivariant message passing by introducing a global diffusion mechanism that maintains equivariance while improving robustness to conformational variation.

## B DIFFUSION PROCESSES ON GRAPHS

This section provides a concise background on diffusion processes defined on graphs. The presentation clarifies how Equation 1 can be interpreted as a graph partial differential equation.

### B.1 DIFFUSION AS FEATURE PROPAGATION

Let $\mathcal{G} = (\mathcal{V}, \mathcal{E})$ be a graph with $n = |\mathcal{V}|$ nodes. A time-dependent node feature field is denoted by

$$\mathbf{Z}(t) = [\mathbf{z}_1(t), \ldots, \mathbf{z}_n(t)]^\top \in \mathbb{R}^{n \times d},$$

where $\mathbf{z}_i(t)$ represents the feature of node $i$ at time $t \geq 0$.

Graph diffusion models describe the evolution of node features through a continuous-time process

$$\frac{\partial \mathbf{Z}(t)}{\partial t} = \mathcal{L}(\mathbf{Z}(t)),$$

where $\mathcal{L}$ is a graph differential operator that governs information propagation across nodes. This formulation generalizes discrete message passing by replacing layer-wise updates with a continuous dynamical system.

### B.2 GRADIENT AND DIVERGENCE ON GRAPHS

Diffusion on graphs can be derived from discrete analogues of gradient and divergence operators.

**Graph gradient.** Given node features $\mathbf{Z} \in \mathbb{R}^{n \times d}$, the gradient operator maps node features to edge features:

$$(\nabla \mathbf{Z})_{ij} = \mathbf{z}_j - \mathbf{z}_i,$$

for node pairs $(i, j)$. This operator measures feature variation across the graph topology.

**Graph divergence.** The divergence operator aggregates edge features back to nodes:

$$(\operatorname{div} \mathbf{F})_i = \sum_j \mathbf{F}_{ij},$$

where $\mathbf{F}_{ij}$ are edge features. It serves as the adjoint of the gradient operator and controls how edge-level information influences node states.

### B.3 DIFFUSION EQUATION ON GRAPHS

Combining gradient and divergence yields the graph diffusion equation

$$\frac{\partial \mathbf{Z}(t)}{\partial t} = \operatorname{div}\left[\mathbf{S}(t) \odot \nabla \mathbf{Z}(t)\right],$$

where $\mathbf{S}(t) \in \mathbb{R}^{n \times n}$ is a diffusivity operator that modulates information flow between node pairs and $\odot$ denotes element-wise interaction between edge fields.

When $\mathbf{S}(t)$ is symmetric and row-stochastic, the diffusion dynamics can be equivalently written as

$$\frac{\partial \mathbf{Z}(t)}{\partial t} = (\mathbf{S}(t) - \mathbf{I})\,\mathbf{Z}(t),$$

which corresponds to a continuous-time random walk on the graph. This formulation shows that diffusion performs feature smoothing while preserving global consistency across nodes.

## C PROOF OF EQUIVARIANCE OF GGND

We prove that the GGND process, as defined by the PDE in Equation ( equation 1) and its rewritten form in Equation ( equation 4), is SE(3)-equivariant. Specifically, the learned features $\mathbf{Z}(T)$ transform correctly under SE(3) transformations (rotations and translations) applied to the input positions $\mathbf{X}$ and invariant scalar features $\mathbf{H}$.

SE(3)-equivariance means that if we apply a transformation $g \in$ SE(3) to the inputs, the output features transform accordingly:

$$\mathbf{Z}'(T) = D(g)\mathbf{Z}(T),$$

where $\mathbf{Z}'(T)$ is the solution of the PDE for the transformed inputs $\mathbf{X}' = g\mathbf{X}$ and $\mathbf{H}' = \mathbf{H}$ (since $\mathbf{H}$ are invariant scalars), and $D(g)$ denotes the group representation acting on the spherical tensor features (irreducible representations of O(3), labeled by $L$ and $M$).

Translations are handled trivially because the model depends only on relative positions $\|\mathbf{x}_{ji}\|$ (invariant) and unit vectors $\hat{\mathbf{x}}_{ji}$ (equivariant under rotations but invariant under translations). The initial embedding $\phi_{\mathcal{E}}(\mathbf{X}, \mathbf{H})$ uses radial basis functions (RBFs) on distances, which are translation-invariant. Thus, the dynamics preserve translation invariance.

We focus on rotation equivariance under $g \in$ SO(3). The representation $D^L(g)$ acts on each irrep component as:

$$\mathbf{z}'_{i,kLM} = \sum_{M'} D^L_{MM'}(g)\mathbf{z}_{i,kLM'},$$

where the action is the same for all nodes $i$.

Assume the initial condition is equivariant: $\mathbf{Z}'(0) = D(g)\mathbf{Z}(0)$. We need to show that if $\mathbf{Z}'(t) = D(g)\mathbf{Z}(t)$ at time $t$, then the time derivative preserves this property:

$$\frac{\partial \mathbf{Z}'(t)}{\partial t} = D(g)\frac{\partial \mathbf{Z}(t)}{\partial t}.$$

This requires showing that the right-hand side operator $f(\mathbf{Z}, \mathbf{X}) = \text{div}\left[\mathbf{S}(\mathbf{Z}(t), \mathbf{X}, t) \odot \nabla \mathbf{Z}(t)\right] = (\mathbf{S} - \mathbf{I})\mathbf{Z}(t)$ is equivariant:

$$f(\mathbf{Z}', \mathbf{X}') = D(g)f(\mathbf{Z}, \mathbf{X}).$$

We prove this by showing that each component—the gradient $\nabla \mathbf{Z}$, the diffusivity $\mathbf{S}$, and their combination—is equivariant.

**Equivariance of the Gradient Operator.** The equivariant gradient is defined as:

$$(\nabla \mathbf{z})_{ij,kl_2m_2} = \sum_{\tilde{k}} W_{k\tilde{k}l_2}(\mathbf{z}_{j,\tilde{k}l_2m_2} - \mathbf{z}_{i,\tilde{k}l_2m_2}),$$

where $W$ are learnable scalar weights (invariant under rotations), and the gradient operates per irrep $L = l_2$ and channel, without changing $L$.

For the transformed features and positions:

$$\mathbf{z}'_{j,\tilde{k}l_2m_2} - \mathbf{z}'_{i,\tilde{k}l_2m_2} = \sum_{m'_2} D^{l_2}_{m_2m'_2}(g)(\mathbf{z}_{j,\tilde{k}l_2m'_2} - \mathbf{z}_{i,\tilde{k}l_2m'_2}).$$

Thus,

$$(\nabla \mathbf{z}')_{ij,kl_2m_2} = \sum_{\tilde{k}} W_{k\tilde{k}l_2} \sum_{m'_2} D^{l_2}_{m_2m'_2}(g)(\mathbf{z}_{j,\tilde{k}l_2m'_2} - \mathbf{z}_{i,\tilde{k}l_2m'_2}) = \sum_{m'_2} D^{l_2}_{m_2m'_2}(g)(\nabla \mathbf{z})_{ij,kl_2m'_2},$$

since $W$ is invariant. The gradient transforms as the same irrep, so $\nabla$ is equivariant: $\nabla \mathbf{Z}' = D(g)(\nabla \mathbf{Z})$.

**Equivariance of the Diffusivity.** The diffusivity (attention matrix) is:

$$\mathbf{S}(t)[i,j]_{kl_3m_3} = \sum_{l_1,l_2,m_1,m_2} C^{l_3m_3}_{l_1m_1,l_2m_2} R_{kl_1l_2l_3}(\|\mathbf{x}_{ji}\|) Y^{l_1}_{m_1}(\hat{\mathbf{x}}_{ji}) \phi(\mathbf{z}_i(t), \mathbf{z}_j(t))_{l_2m_2},$$

where $C$ are Clebsch-Gordan coefficients (invariant), $R$ is a learnable radial function (depends on invariant distance $\|\mathbf{x}_{ji}\|$), $Y^{l_1}_{m_1}$ are spherical harmonics, and $\phi$ is an equivariant pairwise interaction (e.g., gated tensor product).

Under transformation: - $\|\mathbf{x}'_{ji}\| = \|\mathbf{x}_{ji}\|$ (invariant), - $\hat{\mathbf{x}}'_{ji} = g\hat{\mathbf{x}}_{ji}$, and assuming the convention where $Y^{l_1}(\hat{\mathbf{x}}'_{ji}) = \sum_{p_1} D^{l_1}_{m_1p_1}(g)Y^{l_1}_{p_1}(\hat{\mathbf{x}}_{ji})$ (equivariant filter as in NequIP and e3nn), - $\phi(\mathbf{z}'_i, \mathbf{z}'_j)_{l_2m_2} = \sum_{p_2} D^{l_2}_{m_2p_2}(g)\phi(\mathbf{z}_i, \mathbf{z}_j)_{l_2p_2}$ (by assumption that $\phi$ is equivariant).

Substituting:

$$\mathbf{S}'[i,j]_{kl_3m_3} = \sum_{l_1l_2m_1m_2} C^{l_3m_3}_{l_1m_1,l_2m_2} R(\|\mathbf{x}_{ji}\|) \left(\sum_{p_1} D^{l_1}_{m_1p_1}(g)Y^{l_1}_{p_1}(\hat{\mathbf{x}}_{ji})\right) \left(\sum_{p_2} D^{l_2}_{m_2p_2}(g)\phi_{l_2p_2}\right).$$

This is:

$$\sum_{p_1p_2} \sum_{l_1l_2m_1m_2} C^{l_3m_3}_{l_1m_1,l_2m_2} R Y^{l_1}_{p_1} \phi_{l_2p_2} D^{l_1}_{m_1p_1}(g) D^{l_2}_{m_2p_2}(g).$$

Since the tensor product representation is $D^{l_1} \otimes D^{l_2}$, and the CG decomposition to $l_3$ commutes with the group action (CG coefficients are invariant and define an equivariant basis change), the entire expression transforms as the output irrep:

$$\mathbf{S}'[i,j]_{kl_3m_3} = \sum_{m'_3} D^{l_3}_{m_3m'_3}(g)\mathbf{S}[i,j]_{kl_3m'_3}.$$

Thus, $\mathbf{S}(\mathbf{Z}', \mathbf{X}', t) = D(g)\mathbf{S}(\mathbf{Z}, \mathbf{X}, t)D(g)^{-1}$ in the sense of the adjoint action on linear maps, but since we treat $\mathbf{S}$ as producing equivariant filters, the composition preserves equivariance.

**Equivariance of the PDE Operator.** The operator is $f(\mathbf{Z}) = (\mathbf{S} - \mathbf{I})\mathbf{Z}$, which corresponds to $\sum_j \mathbf{S}_{ij} \odot (\mathbf{z}_j - \mathbf{z}_i)$ (assuming $\odot$ is an equivariant bilinear operation, such as channel-wise multiplication or tensor contraction, consistent with the right-stochastic property).

Since $\nabla \mathbf{Z}' = D(g)\nabla \mathbf{Z}$ and $\mathbf{S}' = D(g)\mathbf{S}D(g)^{-1}$ (adjoint for maps), the product $\mathbf{S}' \odot \nabla \mathbf{Z}' = D(g)(\mathbf{S} \odot \nabla \mathbf{Z})$, because:

$$D(g)\mathbf{S}D(g)^{-1} \odot D(g)(\nabla \mathbf{Z}) = D(g)(\mathbf{S} \odot \nabla \mathbf{Z}),$$

assuming $\odot$ commutes with $D(g)$ (as it does for tensor products or contractions in irrep bases).

The divergence $\operatorname{div}$ is a sum over $j$, which is permutation-invariant and commutes with $D(g)$:

$$\operatorname{div}[\mathbf{S}' \odot \nabla \mathbf{Z}'] = \sum_j D(g)(\mathbf{S}_{ij} \odot (\mathbf{z}_j - \mathbf{z}_i)) = D(g)\operatorname{div}[\mathbf{S} \odot \nabla \mathbf{Z}].$$

Thus, $f(\mathbf{Z}', \mathbf{X}') = D(g)f(\mathbf{Z}, \mathbf{X})$.

Since the operator is equivariant, the solution to the PDE (e.g., via numerical integration like Euler steps) preserves equivariance: $\mathbf{Z}'(t) = D(g)\mathbf{Z}(t)$ for all $t$, including $t = T$.

Finally, the energy prediction uses the invariant components $\mathbf{z}_{i,k00}(T)$ ($L = 0$, scalar invariants), which are unchanged under $D(g)$, ensuring the total energy $E_{\mathrm{GGND}}$ is SE(3)-invariant.

## D PROOFS OF REGRET BOUND

**Proof of Equation (8).** Assume that the loss function $\ell$ and the decoder $\phi_{\mathrm{dec}}$ are Lipschitz continuous with constants $L_1$ and $L_2$, respectively. According to the topology formation hypothesis in Section 2, we decompose the joint distribution as $p(X, H, A, Y \mid E, M) = p(X, H \mid E, M)p(A, Y \mid X, H, E, M)$. Since $X = f(U; W)$ and $H = h(U; W)$ are independent of $E$ and $M$, it follows that $p(X, H \mid E_{\mathrm{tr}}, M_{\mathrm{tr}}) = p(X, H \mid E_{\mathrm{te}}, M_{\mathrm{tr}})$.

The extrapolation gap is:

(↑)
$$\begin{aligned}
|\mathcal{L}(\Gamma_\theta; E_{\mathrm{te}}, M_{\mathrm{tr}}) - \mathcal{L}_{\mathrm{tr}}(\Gamma_\theta; E_{\mathrm{tr}}, M_{\mathrm{tr}})| &\leq |\mathcal{L}(\Gamma_\theta; E_{\mathrm{te}}, M_{\mathrm{tr}}) - \mathcal{L}(\Gamma_\theta; E_{\mathrm{tr}}, M_{\mathrm{tr}})| \\
&+ |\mathcal{L}(\Gamma_\theta; E_{\mathrm{tr}}, M_{\mathrm{tr}}) - \mathcal{L}_{\mathrm{tr}}(\Gamma_\theta; E_{\mathrm{tr}}, M_{\mathrm{tr}})|.
\end{aligned} \tag{9}$$

The second term is bounded by $\mathcal{D}_{\mathrm{in}}(\Gamma_\theta, E_{\mathrm{tr}}, M_{\mathrm{tr}}, N_{\mathrm{tr}})$ with probability at least $1 - \delta$ by standard extrapolation bounds (Equation (7)).

For the first term:

$$\begin{aligned}
\mathcal{L}(\Gamma_\theta; E_{\mathrm{te}}, M_{\mathrm{tr}}) - \mathcal{L}(\Gamma_\theta; E_{\mathrm{tr}}, M_{\mathrm{tr}}) &= \mathbb{E}_{X', H', A', Y' \sim p(\cdot | E_{\mathrm{te}}, M_{\mathrm{tr}})}[\ell(\Gamma_\theta(X', H', A'), Y')] \\
&- \mathbb{E}_{X, H, A, Y \sim p(\cdot | E_{\mathrm{tr}}, M_{\mathrm{tr}})}[\ell(\Gamma_\theta(X, H, A), Y)].
\end{aligned} \tag{10}$$

Since the marginals over $X, H$ are identical:

$$= \mathbb{E}_{X,H}\left[\mathbb{E}_{A',Y'|X,H}[\ell(\Gamma_\theta(X, H, A'), Y')] - \mathbb{E}_{A,Y|X,H}[\ell(\Gamma_\theta(X, H, A), Y)]\right]. \tag{11}$$

By the triangle inequality and Lipschitz continuity of $\ell$:

$$\begin{aligned}
&\leq \mathbb{E}_{X,H,A,A',Y'}|\ell(\Gamma_\theta(X, H, A'), Y') - \ell(\Gamma_\theta(X, H, A), Y')| \\
&+ \mathbb{E}_{X,H,A,Y,Y'}|\ell(\Gamma_\theta(X, H, A), Y') - \ell(\Gamma_\theta(X, H, A), Y)|,
\end{aligned} \tag{12}$$

where $A, Y \sim p(\cdot \mid E_{\mathrm{tr}}, M_{\mathrm{tr}})$ and $A', Y' \sim p(\cdot \mid E_{\mathrm{te}}, M_{\mathrm{tr}})$. Applying Lipschitz constants:

$$\leq L_1 \cdot \mathbb{E}\|\mathbf{Z}(T; A') - \mathbf{Z}(T; A)\|_2 + L_2 \cdot \mathbb{E}\|\mathbf{Y}' - \mathbf{Y}\|_2,$$

yielding the decomposition after rescaling constants into $\mathcal{O}(\cdot)$.

**Proof of Theorem 3.1.** The equivariant graph neural diffusion model assumes full connectivity, with diffusivity $\mathbf{S}(t)$ defined based on positions $\mathbf{X}$ and features $\mathbf{Z}(t)$ (Equation (3)), independent of the adjacency $\mathbf{A}$. Thus, the solution $\mathbf{Z}(T)$ to Equation (4) does not depend on $\mathbf{A}$, implying $\|\mathbf{Z}(T; \mathbf{A}') - \mathbf{Z}(T; \mathbf{A})\|_2 = 0$.

Since $0 = \mathcal{O}(\psi(\|\Delta\tilde{\mathbf{A}}\|_2))$ for any arbitrary polynomial function $\psi$, the variation magnitude is reduced to any order. The injectivity of $f$ and $h$ ensures that latent variables $U$ can be recovered from $\mathbf{X}$ and $\mathbf{H}$, enabling the equivariant attention in $\mathbf{S}(t)$ to capture latent interactions from the graphon $W$, independent of shifts in $\mathbf{A}$.

**Proof of Corollary 3.2.** The conclusion follows directly by substituting the result of Equation (8) into the $\mathcal{D}_{\mathrm{M}}$.

## E   PARAMETER

The training details are outlined below, with dataset-specific parameters provided in Table 4.

Table 4: Dataset Information and Dataset-specific Parameters

| Dataset | 3BPA | SAMD23 | |
|---|---|---|---|
| Molecule | 3BPA | SiN | HfO |
| Atoms | 27 | 16-510 | 96 |
| Batch size | 4 | 1 | 2 |
| Epochs | 1,000 | 200 | 200 |
| Training Time (h) | 4 | 30 | 24 |

**Training**

1. Optimizer: Adam (Kingma & Ba, 2015) optimizer is used with a constant learning rate of $10^{-4}$ as our default training configuration.
2. GPU: NVIDIA GeForce RTX 3090
3. CPU: Intel(R) Xeon(R) Platinum 8338C CPU
4. Memory: 512 GB

For baselines, we adopt the recommended parameters from their original publications. Specifically, for the message-passing layer in our implementations with GGND, we include an additional layer for geometric graph neural diffusion. To ensure a fair comparison with similar model capacity, we reduce one message-passing layer in the baselines during integration.

## F   COMPUTATIONAL OVERHEAD ANALYSIS AND COMPARISON

Table 5 summarizes the computational and memory overhead incurred when integrating GGND into ViSNet on the 3BPA dataset. All models are trained using identical hyperparameters; ViSNet* includes an additional 500 training epochs to provide a more comprehensive point of comparison. Experiments were conducted on an NVIDIA GeForce RTX 3090 GPU paired with an Intel(R) Xeon(R) Platinum 8338C CPU.

Overall, incorporating GGND increases training time by 26.54% and MD simulation time by 15.57%. GPU memory consumption increases by 15.28% during training and 14.87% during inference. Considering the substantial gains in energy/force accuracy and stability, these additional computational and memory costs are acceptable in practice.

## G   DATASETS

The 3BPA dataset consists of configurations of the flexible, drug-like molecule 3-(benzyloxy) pyridin-2-amine. Initial configurations were generated from short (0.5 ps) molecular dynamics sim-

Table 5: Computational and Memory Overhead Introduced by GGND

|  | Metrics | VisNet | VisNet* | +GGND |
|---|---|---|---|---|
| Time | Training Time (h) | 6.822 | 10.326 | 8.633 |
|  | Inference Time (s) | 13.872 | 13.672 | 16.032 |
|  | MD Time for 100 ps (h) | 1.958 | 1.949 | 2.282 |
| Memory | Training Memory | 20.457 | 20.455 | 23.582 |
|  | Inference Memory (GiB) | 14.125 | 14.248 | 16.225 |

∗: ViSNet* includes an additional 500 training epochs.

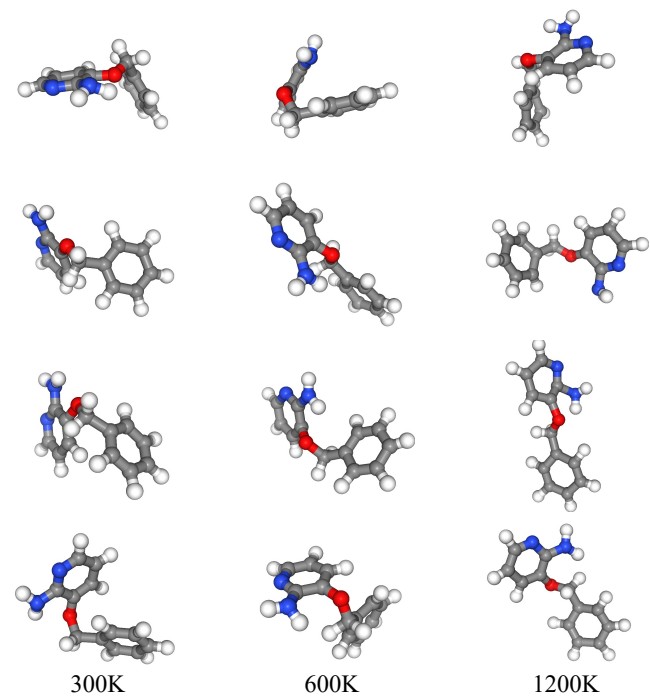

300K       600K       1200K

Figure 5: Representative 3BPA Molecular Configurations Sampled from MD at 300, 600, and 1200 K.

ulations using the ANI-1x force field to bias sampling toward lower-energy regions of the potential energy surface. In addition, longer 25 ps MD simulations were performed at three temperatures—300, 600, and 1200 K—using a Langevin thermostat with a 1 fs time step. A selection of these configurations is visualized in Figure 5.

## H MD SIMULATION UNDER NVE AND NVT ENSEMBLES

We perform constant-energy (NVE) and constant-temperature (NVT) molecular dynamics simulations at 1200 K on the 3BPA dataset. We use Velocity-Verlet integration for 200 ps with a 1 fs timestep for the NVE ensembles. We perform Langevin dynamics at a temperature of 1200K, a timestep of 1.0 fs, and a friction coefficient of 0.01 fs$^{-1}$, for 200,000 steps, corresponding to 200 ps for NVT ensemble.

In both the NVE and NVT molecular dynamics simulations, GGND demonstrates markedly superior stability compared to MACE and VisNet (Figures 6 and 7). GGND achieves the lowest Averaged Max Bond Length Deviation across the full 200 ps trajectories and maintains stable geometries, with only a few isolated instances where the Max Bond Length Deviation slightly exceeds the threshold. In contrast, MACE begins to violate the bond-length threshold at approximately 13 ps in the NVE

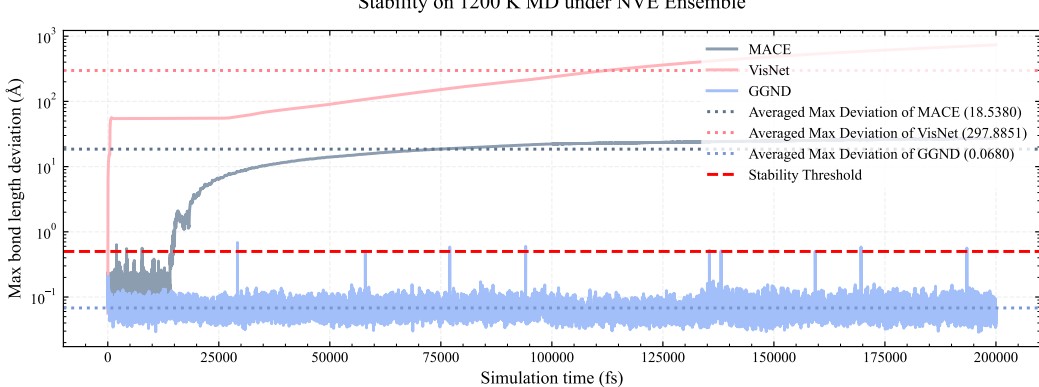

Figure 6: Stability of MD Simulations under NVE Ensemble on 3BPA.

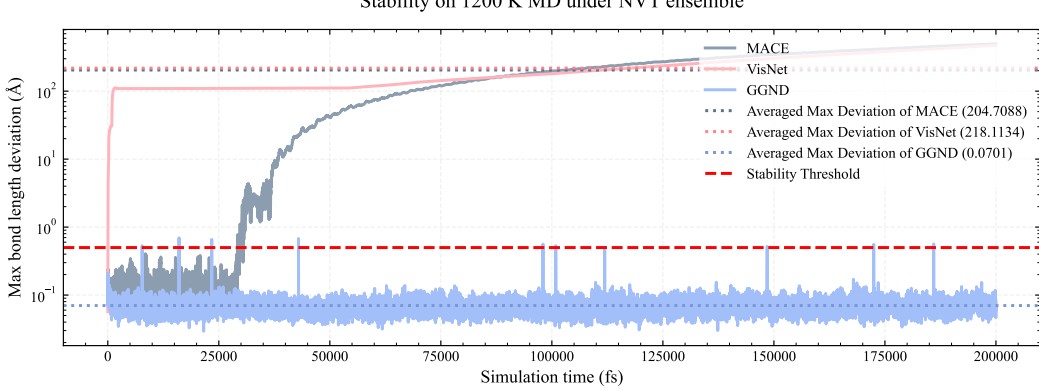

Figure 7: Stability of MD Simulations under NVT Ensemble on 3BPA.

ensemble and around 27 ps in the NVT ensemble, after which the deviations steadily grow. Vis-Net performs even worse, showing threshold-breaking behavior almost immediately and displaying rapidly increasing deviations throughout the simulation. Overall, these results highlight the robustness and stability of GGND in unseen temperature MD settings.

# I    LIMITATIONS

Although the GGND framework demonstrates strong performance in enhancing stability and accuracy for small to medium-sized molecular systems, such as those in the 3BPA and SAMD23 datasets with up to 510 atoms, its scalability to larger biomolecular systems like proteins comprising millions of atoms remains a critical limitation. The method's reliance on a fully-connected graph for the diffusion process, which facilitates all-pair information flows, introduces quadratic computational complexity in both time and memory with respect to the number of atoms, rendering it impractical for real-world applications involving extensive simulations. Future developments could incorporate sparse approximations or hierarchical diffusion mechanisms to mitigate these issues and extend GGND's utility to large-scale protein dynamics.

# J    IMPACT STATEMENTS

This paper presents research aimed at advancing Artificial Intelligence (AI) applications in scientific domains, including materials science, chemistry, and biology. The insights and expertise gained will significantly enhance AI technologies, accelerating the process of scientific discovery.

Machine learning for molecular dynamics enables rapid molecular analysis. However, the potential for misuse and unintended consequences underscores the need for stringent ethical guidelines, robust regulations, and responsible deployment to safeguard individuals and society from harm.

## K   THE USE OF LARGE LANGUAGE MODELS

The core method development and research ideation in this paper were conducted independently of LLMs, and LLMs did not contribute to any original or non-standard components of the work. The authors utilized LLMs solely as a general-purpose assist tool for checking grammar and improving the clarity of the manuscript, as well as for aiding in the comprehension of existing literature. All content in this submission, including any text refined with LLM assistance, has been thoroughly reviewed by the authors, who take full responsibility for its accuracy, integrity, and compliance with ethical standards. No LLMs are considered contributors or eligible for authorship.

