# OpenReview forum: "Geometric Graph Neural Diffusion for Stable Molecular Dynamics Simulations"
_ICLR.cc/2026/Conference — ICLR 2026 Poster_

### Official Review · Reviewer_Ev5w · 2025-10-28

**Soundness:** 2
**Presentation:** 2
**Contribution:** 2
**Rating:** 2
**Confidence:** 5

**Summary:**

The study aim to address a critical limitation of existing Geometric Graph Neural Networks (Geo-GNNs) in molecular dynamics (MD) simulations: minor force prediction errors can destabilize long-temporal trajectories, especially when extrapolating to unseen molecular conformations (e.g., different temperatures) due to limited training data coverage. To solve this, the authors propose GGND, a plug-and-play framework that captures geometrically invariant topological features via an equivariant diffusion process on fully connected graphs, mitigating error accumulation and ensuring stable MD simulations. Experiments were conducted on the 3BPA and SAMD23 datasetsshow that GGND outperforms baselines in both accuracy and stability.

**Strengths:**

1. The paper provides solid theoretical support: it formalizes the causal mechanism of geometric topology formation (linking environment, cutoff radius, and latent graphons) and derives a regret bound for extrapolation error under topological shifts. Importantly, it proves GGND’s SE(3) equivariance via detailed analysis of gradient/diffusivity operators
2.  It compares with 4 representative baselines (NequIP, MACE, SEGNO, VisNet) and shows its usefulness in different models.

**Weaknesses:**

1. Scalability Limitation for Large Biomolecular Systems​
GGND’s fully connected graph diffusion introduces quadratic computational complexity (O(N²) for N atoms), making it impractical for large systems like proteins (millions of atoms) or complex biomolecules. The paper only tests systems up to 510 atoms (SAMD23 HfO), and its memory/time costs will grow exponentially with larger N. This restricts its real-world applicability to small/medium molecules, as large-scale MD simulations (e.g., protein folding) are common in biology and drug discovery.
2. Incomplete stability analysis: While bond length and RDF are used to measure stability, the paper does not evaluate other critical MD metrics (e.g., energy conservation, pressure fluctuations, or thermodynamic observables like melting points), which are essential for validating real-world utility.
3. Unclear relationship between diffusion operation and geometric topology. It lacks illustration how the diffusion operations map diffusion structural topology.
4. Lack strcuture analysis. Simulating a biomolecule at 1200K is WEIRD and meanningless. You can get nothing from the unrealistic experiment. The author should carefully check the structure changes during the simulation or perform another one under a normal environment.
5. lack interpretability: while it mentions attention weights reflect interaction strength, no visualizations are provided to illustrate how it captures unobserved topological features.
6. Time consumption
Again, the diffusion module seems ultra time-consuming. The authors should seriously analyze and demonstrate the extra huge computational cost during model training compared to the baseline model, e.g., ViSNet. Will it be applied to large molecular systems?
7. Misleading concept of Generalization
The defination in this article seems a little misleading. When taking about generalization, the generalizable ability among different kinds of molecules, i.e., model trained in a molecules and tested on other kinds of molecules is much more important and convincing than that for different conformations of the same kind of molecule. Unfortunately, through the whole article, the authors neglected the important point for some reason.
8. Unphysical structures
Again, it's so werid and totally meaningless to sample protein structures at such high temerature (1200K). First, the author must show the structes they sampled at that temperature. I guess most of structures are totally collapsed. Second, the authors should perform classical MD simulations at such meaningless temperature to compare the RMSD/system energy or other observables with the models shown in the article.

**Questions:**

The authors should seriously address the concerns shown in Weakness point by point to improve the quality of the paper.

---

> ### Author Response · Authors · 2025-11-21
> **First Response to Reviewer Ev5w [1/2]**
>
> # Response to Reviewer Ev5w
> We appreciate the reviewer for the time and effort on reviewing.
> # Response to Weakness 1
> > Scalability Limitation for Large Biomolecular Systems GGND’s fully connected graph diffusion introduces quadratic computational complexity (O(N²) for N atoms), making it impractical for large systems like proteins (millions of atoms) or complex biomolecules. The paper only tests systems up to 510 atoms (SAMD23 HfO), and its memory/time costs will grow exponentially with larger N. This restricts its real-world applicability to small/medium molecules, as large-scale MD simulations (e.g., protein folding) are common in biology and drug discovery.
>
> We respectively agree with the reviewer that scalability is a critical aspect. We have discussed this limitation in Section F of the Appendix. We want to highlight that the focus of this paper is on improving the stability, which is equally important for real-world applications [1, 2]. We will leave the scalability to our future work.
>
> [1] Fu, Xiang, et al. "Forces are not Enough: Benchmark and Critical Evaluation for Machine Learning Force Fields with Molecular Simulations." Transactions on Machine Learning Research.
>
> [2] Frank, J. Thorben, et al. "A Euclidean transformer for fast and stable machine learned force fields." Nature Communications 15.1 (2024): 6539.
>
>
> # Response to Weakness 2
> > Incomplete stability analysis: While bond length and RDF are used to measure stability, the paper does not evaluate other critical MD metrics (e.g., energy conservation, pressure fluctuations, or thermodynamic observables like melting points), which are essential for validating real-world utility.
>
> We appreciate this important point. We use these stability metrics following the existing works for fair comparison [1,2,3]. We fully agree that evaluating additional thermodynamic and dynamical observables would provide a more comprehensive assessment. Expanding the stability benchmark along these dimensions is an important direction, and we plan to investigate this in future work.
>
>
> [1] Fu, Xiang, et al. "Forces are not Enough: Benchmark and Critical Evaluation for Machine Learning Force Fields with Molecular Simulations." Transactions on Machine Learning Research.
>
> [2] Amin, Ishan, Sanjeev Raja, and Aditi S. Krishnapriyan. "Towards Fast, Specialized Machine Learning Force Fields: Distilling Foundation Models via Energy Hessians." The Thirteenth International Conference on Learning Representations.
>
> [3] Kim, Geonu, et al. "Benchmark of machine learning force fields for semiconductor simulations: datasets, metrics, and comparative analysis." Advances in neural information processing systems 36 (2023): 51434-51476.
>
> # Response to Weakness 3
> > Unclear relationship between diffusion operation and geometric topology. It lacks illustration how the diffusion operations map diffusion structural topology.
>
> The diffusion operators act on the fully-connected geometric graph topology to capture the geometrically invariant topological features.
> Specifically, the diffusion operators---The gradient operator maps node fields to edge fields, while the divergence operator div is its adjoint, mapping edge fields back to nodes. Together, diffusion operations perform global message passing. As for the cutoff-based geometric topology, we employ a local EGNN to capture the local environment of the atomic system. The ablation study shows that the diffusion on local geometric topology (with a predefined cutoff) is incapable of addressing the stability.
>
> # Response to Weakness 4
> > Lack strcuture analysis. Simulating a biomolecule at 1200K is WEIRD and meanningless. You can get nothing from the unrealistic experiment. The author should carefully check the structure changes during the simulation or perform another one under a normal environment.
>
> The dataset 3BPA is well accepted in many papers to evaluate the OOD generalization [1, 2, 3]. We agree that 1200 K may not be a physically realistic biological temperature. Our focus is not to model protein behavior at 1200 K but to investigate the OOD perturbations that test stability under geometric shifts.
>
> [1] Batatia, Ilyes, et al. "MACE: Higher order equivariant message passing neural networks for fast and accurate force fields." Advances in neural information processing systems 35 (2022): 11423-11436.
>
> [2] Kovács, Dávid Péter, et al. "Linear atomic cluster expansion force fields for organic molecules: beyond rmse." Journal of chemical theory and computation 17.12 (2021): 7696-7711.
>
> [3] Musaelian, Albert, et al. "Learning local equivariant representations for large-scale atomistic dynamics." Nature Communications 14.1 (2023): 579.

---

> ### Author Response · Authors · 2025-11-21
> **First Response to Reviewer Ev5w [2/2]**
>
> # Response to Weakness 5
> > lack interpretability: while it mentions attention weights reflect interaction strength, no visualizations are provided to illustrate how it captures unobserved topological features.
>
> We acknowledge that interpretability is an important topic, and it is out of the scope of our paper.
>
> As for the attention weights, we did not state that the ```attention weights reflect interaction strength```. The attention weights are learned to capture latent interactions that are independent of shifts in the adjacency matrix $\mathbf{A}$. This allows the model to implicitly capture interaction patterns and topological relations that may not be explicitly observed in the training molecular graphs. We hope that this clarification provides a more concrete intuition for how GGND represents unobserved topological features.
>
> # Response to Weakness 6
> > Time consumption Again, the diffusion module seems ultra time-consuming. The authors should seriously analyze and demonstrate the extra huge computational cost during model training compared to the baseline model, e.g., ViSNet. Will it be applied to large molecular systems?
>
> We summarize the computational and memory overhead incurred when integrating GGND into ViSNet in the Table below. All models are trained using identical hyperparameters; ViSNet* includes an additional 500 training epochs to provide a more comprehensive point of comparison. Experiments were conducted on an NVIDIA GeForce RTX 3090 GPU paired with an Intel(R) Xeon(R) Platinum 8338C CPU.
>
> Overall, incorporating GGND increases training time by 26.54\% and MD simulation time by 15.57\%. GPU memory consumption increases by 15.28\% during training and 14.87\% during inference. Considering the substantial gains in energy/force accuracy and stability, these additional computational and memory costs are acceptable in practice.
>
> Again, we acknowledge the challenge of scaling to extremely large atomic systems. Nevertheless, GGND has been demonstrated to perform effectively on the SAMD23 dataset (up to 510 atoms), which contains  comparatively large atomic systems.
>
> | | Metrics | VisNet | VisNet* | +GGND |
> | - | - | - | - | - |
> | Epoch | | 1000 | 1500 | 1000 |
> | Time | Training Time (h) | 6.822 | 10.326 | 8.633 |
> | | Inference Time (s) | 13.872 | 13.672 | 16.032 |
> | | MD Time for 100 ps (h) | 1.358 | 1.349 | 1.582 |
> | Space | Training Memory (GiB) | 20.457 | 20.455 | 23.582 |
> | | Inference Memory (GiB) | 14.125 | 14.248 | 16.225 |
> | 1200K | Enery (eV/Å) | 3.464 | 3.193 | 0.583 |
> | | Forces (eV/A) | 1.404 | 1.383 | 0.304 |
> | | Stability (ps) | 0.004 | 0.009 | 11.209 |
>
> # Response to Weakness 7
> > Misleading concept of Generalization The defination in this article seems a little misleading. When taking about generalization, the generalizable ability among different kinds of molecules, i.e., model trained in a molecules and tested on other kinds of molecules is much more important and convincing than that for different conformations of the same kind of molecule. Unfortunately, through the whole article, the authors neglected the important point for some reason.
>
> We would like to highlight that generalization is a very large topic and contains many aspects, even in molecular dynamics simulations. Generalizing to different kinds of molecules is only one type of generalization task; there are many papers that discuss the generalization to different environments [1-4].
>
> [1] Batatia, Ilyes, et al. "MACE: Higher order equivariant message passing neural networks for fast and accurate force fields." Advances in neural information processing systems 35 (2022): 11423-11436.
>
> [2] Kovács, Dávid Péter, et al. "Linear atomic cluster expansion force fields for organic molecules: beyond rmse." Journal of chemical theory and computation 17.12 (2021): 7696-7711.
>
> [3] Musaelian, Albert, et al. "Learning local equivariant representations for large-scale atomistic dynamics." Nature Communications 14.1 (2023): 579.
>
> [4] Yang, Han, et al. "Mattersim: A deep learning atomistic model across elements, temperatures and pressures." arXiv preprint arXiv:2405.04967 (2024).
>
> # Response to Weakness 8
> > Unphysical structures Again, it's so werid and totally meaningless to sample protein structures at such high temerature (1200K). First, the author must show the structes they sampled at that temperature. I guess most of structures are totally collapsed. Second, the authors should perform classical MD simulations at such meaningless temperature to compare the RMSD/system energy or other observables with the models shown in the article.
>
> We have visualized these conformations in Figure 5, Section D of the revised paper. We also added classical MD baselines at the same temperature in Figure 4 of the revised submission.

---

### Official Review · Reviewer_Xkok · 2025-11-01

**Soundness:** 3
**Presentation:** 3
**Contribution:** 3
**Rating:** 8
**Confidence:** 4

**Summary:**

The paper addresses a critical instability issue in Geometric Graph Neural Networks (Geo-GNNs) used for molecular dynamics (MD) simulations. It identifies the cause as "geometric topological shifts," where models trained on certain molecular conformations (e.g., at 300 K) fail to extrapolate to unseen conformations (e.g., at 1200 K).

The authors propose Geometric Graph Neural Diffusion (GGND), a plug-and-play module that runs an equivariant diffusion process on a fully-connected graph. This allows the model to capture global, all-pair atomic information, making it robust to the local topological changes that cause standard models to fail. The paper provides a theoretical regret bound to justify this improved robustness and shows strong empirical results, with GGND massively improving simulation stability on the 3BPA and SAMD23 datasets.

**Strengths:**

**Significance**: It tackles a major practical bottleneck for ML force fields: long-term simulation stability. Solving this OOD extrapolation problem is critical for reliable MD simulations.

**Originality**: The solution is novel. Augmenting local message passing with a parallel, global equivariant diffusion process is a creative and technically sound approach to capturing physics robustly.

**Quality**: The paper is high-quality. The theoretical analysis provides a formal justification for the method's robustness. The experiments are excellent, using temperature variations (300 K vs 1200 K) to create a perfect test for the hypothesis, and the results are dramatic and convincing.

**Clarity**: The paper is well-written, clearly motivating the problem as a "geometric topological shift" and explaining the GGND solution .

**Weaknesses:**

**Scalability**: This is the most significant weakness. The method's reliance on a fully-connected graph introduces $O(N^2)$ computational and memory complexity, which the authors admit is "impractical" for the large-scale systems (e.g., proteins) where this stability is most needed.

**Questions:**

**Scalability**: Given the $O(N^2)$ bottleneck, have you explored approximations? For example, what is the performance trade-off if you use a large-radius cutoff (e.g., 20-30 Å) instead of a fully-connected graph?

**Computational Cost**: What is the practical wall-clock time overhead for a simulation step when adding the GGND module to a baseline like VisNet?

---

> ### Author Response · Authors · 2025-11-21
> **First Response to Reviewer Xkok**
>
> # Response to Reviewer Xkok
> We express our sincere gratitude for the reviewer’s acknowledgment of the strengths inherent in our work. We are delighted that the reviewer identified our method as significant and novel, and that the paper is well-written and of high quality.
> # Response to Weakness 1 and Question 1
> > Scalability: This is the most significant weakness. The method's reliance on a fully-connected graph introduces  computational and memory complexity, which the authors admit is "impractical" for the large-scale systems (e.g., proteins) where this stability is most needed.
> > Scalability: Given the  bottleneck, have you explored approximations? For example, what is the performance trade-off if you use a large-radius cutoff (e.g., 20-30 Å) instead of a fully-connected graph?
>
> We appreciate the reviewer's insightful comment. The current implementation of GGND uses a dense diffusion operator with $O(N^2)$ time and memory. This indeed becomes a bottleneck for large biomolecular systems with thousands of atoms. We acknowledge the challenge of achieving scalability and stability in one framework. We would like to investigate the scalable and stable MD methods in our future work.
>
> On the one hand, the experiments show that the computational complexity is acceptable on the SAMD23 dataset, which contains molecules with up to 510 atoms. Our algorithm shows comparative performance and acceptable time consumption.
>
> On the other hand, as far as we know, there are no available datasets that contain large molecules (radius largely greater than 20-30 Å) while sampled from different environments. However, to alleviate the reviewer's concern, we would like to highlight that the ablation study has shown that the diffusion on the local graph is not satisfied. We hope these discussions could alleviate the reviewer's concern.
>
> # Response to Question 2
> > Computational Cost: What is the practical wall-clock time overhead for a simulation step when adding the GGND module to a baseline like VisNet?
>
> We appreciate this important question. We report below the wall-clock time overhead for training, inference, and MD simulation when adding GGND to ViSNet. All methods use identical hyperparameters; additionally, ViSNet* includes 500 extra training epochs to enable a fairer comparison.
>
> Overall, GGND increases training time by 26.54% and MD simulation time by 16.58%, with a per-step simulation overhead of approximately **0.0117 s**. Given the substantial gains in energy/force accuracy and the several orders of magnitude improvement in stability, we believe this additional computational cost is practical.
>
> | | Metrics | VisNet | VisNet* | +GGND |
> | - | - | - | - | - |
> | Epoch | | 1000 | 1500 | 1000 |
> | Time | Training Time (h) | 6.822 | 10.326 | 8.633 |
> | | Inference Time (s) | 13.872 | 13.672 | 16.032 |
> | | MD Time for 100 ps (h) | 1.358 | 1.349 | 1.582 |
> | Space | Training Memory (GiB) | 20.457 | 20.455 | 23.582 |
> | | Inference Memory (GiB) | 14.125 | 14.248 | 16.225 |
> | 1200K | Enery (eV/Å) | 3.464 | 3.193 | 0.583 |
> | | Forces (eV/A) | 1.404 | 1.383 | 0.304 |
> | | Stability (ps) | 0.004 | 0.009 | 11.209 |

---

### Official Review · Reviewer_WP7J · 2025-11-01

**Soundness:** 4
**Presentation:** 3
**Contribution:** 4
**Rating:** 8
**Confidence:** 3

**Summary:**

This paper addresses a critical challenge for Geometric Graph Neural Networks (Geo-GNNs) in molecular dynamics (MD) simulations: minor prediction errors can accumulate and destabilize long-term MD trajectories. The authors attribute this problem to the model's poor ability to extrapolate to unseen conformations, particularly those arising from environmental changes like different temperatures, a phenomenon they formalize as "Geometric Topological Shifts".

To tackle this, the paper proposes a novel framework called Geometric Graph Neural Diffusion (GGND). The core idea of GGND is to serve as a "plug-and-play" module that integrates with existing local equivariant message-passing frameworks. It achieves this by performing an equivariant diffusion process on a fully-connected graph, enabling instantaneous information flow between arbitrary atomic pairs. This process utilizes novel "equivariant gradient" and "equivariant diffusivity" operators, designed to capture geometrically invariant topological features, thereby mitigating error accumulation and ensuring stable MD simulations.

**Strengths:**

### Novel and Effective Method:
The GGND framework is a novel contribution. Its core design—applying equivariant diffusion on a fully-connected graph —is intuitively sound. It overcomes the limitations of local message passing during topological changes by capturing "all-pair information flows". The ablation study  effectively demonstrates the superiority of "fully-connected diffusion" over "local diffusion" and "fully-connected message passing," validating the design's efficacy.

### Sufficient Experimental Validation:
- The experiment on the 3BPA dataset is well-designed, using different temperatures (300K, 600K, 1200K) to simulate conformational shifts. The results are impressive; for example, at 1200K, VisNet's stability is boosted from 0.004 ps to 11.209 ps, and MACE's stability also sees a 15-fold improvement.

- The comparison against multiple SOTA models on the SAMD23 dataset shows that GGND not only enhances stability but is also highly competitive in force and energy prediction accuracy on OOD splits.

- Solid Theoretical Support: The authors provide a theoretical analysis for GGND, including a proof of SE(3) equivariance and a regret bound under geometric topological shifts , which adds to the work's rigor.

### Strong Practicality:
GGND is designed as a plug-and-play module that can be seamlessly integrated into various existing equivariant GNN architectures. This significantly increases the method's potential applicability and impact.

**Weaknesses:**

### Scalability:
This is the paper's most significant limitation. The method's reliance on diffusion over a fully-connected graph incurs $O(N^2)$ computational and memory complexity (where N is the number of atoms). The authors acknowledge this makes the method "impractical" for large biomolecular systems. While the SAMD23 dataset reaches up to 510 atoms, this is still orders of magnitude smaller than many common systems in MD simulations

### Justification for the Diffusion Framework:
The ablation study shows that fully-connected diffusion (GGND) outperforms a fully-connected message passing baseline (GGND‡). The paper attributes the baseline's poorer performance to "training challenges". This explanation is somewhat brief and does not fully explore why the PDE-based diffusion process is inherently superior to a standard (but also fully-connected) message-passing layer for this task.

**Questions:**

1. Scalability of GGND:

While the proposed method shows significant improvements in smaller molecular systems, can GGND scale efficiently to larger molecules or systems with a higher number of atoms? What are the computational bottlenecks, and how might they be addressed?

2. Comparison to Other Approaches:

How does GGND compare with other domain adaptation methods, such as active learning or test-time adaptation, in terms of improving stability and extrapolation? Have these methods been considered as part of your baseline comparisons?

3. Model Limitations:

The theoretical analysis shows GGND's ability to manage geometric topological shifts. Are there any specific types of conformational shifts or molecular behaviors that GGND might still struggle with, even with its diffusion mechanism?

**Details Of Ethics Concerns:**

No Ethics Concerns

---

> ### Author Response · Authors · 2025-11-21
> **First Response to Reviewer WP7J**
>
> # Response to Reviewer WP7J
> We appreciate the reviewer’s recognition of the strengths in our work. We are glad that the reviewer found that our method is novel and effective, our experiments are sufficient, and the design is practical.
> # Response to Weakness 1 and Question 1
> > Scalability:
> > This is the paper's most significant limitation. The method's reliance on diffusion over a fully-connected graph incurs  computational and memory complexity (where N is the number of atoms). The authors acknowledge this makes the method "impractical" for large biomolecular systems. While the SAMD23 dataset reaches up to 510 atoms, this is still orders of magnitude smaller than many common systems in MD simulations
> > Scalability of GGND:
> > While the proposed method shows significant improvements in smaller molecular systems, can GGND scale efficiently to larger molecules or systems with a higher number of atoms? What are the computational bottlenecks, and how might they be addressed?
>
> We fully agree that scalability is a key limitation of the current instantiation of GGND. We acknowledge that 510 atoms is not large in MD simulations, but this scale has already been recognized as comparatively large in existing ML methods for MD simulations.
>
> The computational bottlenecks arise from applying a diffusion operator over a complete latent graph; we would like to explore applying structured sparsification, hierarchical diffusion, or low-rank / Nyström approximations in our future work to improve the scalability of our work.
>
> # Response to Weakness 2
> > Justification for the Diffusion Framework:
> > The ablation study shows that fully-connected diffusion (GGND) outperforms a fully-connected message passing baseline (GGND‡). The paper attributes the baseline's poorer performance to "training challenges". This explanation is somewhat brief and does not fully explore why the PDE-based diffusion process is inherently superior to a standard (but also fully-connected) message-passing layer for this task.
>
> We agree that the advantage of diffusion over a fully-connected message passing baseline deserves a more rigorous explanation. At its core, the benefit of GGND’s PDE-style diffusion is that it mitigates oversmoothing and remains stable even with many diffusion “layers”, unlike a naive fully-connected MPNN. This is directly inspired by and analogous to the behavior of Graph Neural.
>
> Specifically, the diffusion formulation imposes an inductive bias: the model’s time evolution follows a PDE with known, smooth dynamics. This structure regularizes learning, making it easier to train than a fully-connected, unconstrained MPNN, which may struggle with gradient instability when depth and connectivity are high. Besides, the continuous diffusion formulation can be integrated stably (using explicit or implicit solvers) over many steps, allowing arbitrarily many “layers” (i.e., diffusion steps).
>
> # Response to Question 2
> > Comparison to Other Approaches:
> How does GGND compare with other domain adaptation methods, such as active learning or test-time adaptation, in terms of improving stability and extrapolation? Have these methods been considered as part of your baseline comparisons?
>
> We appreciate this insightful comment. In our initial submission, we have incorporated SEGN as a representative OOD generalization baseline. Regarding test-time adaptation or active learning approaches: although these techniques are indeed promising for improving stability and extrapolation, they typically require additional MD data or online data collection, which makes them difficult to compare fairly within our current experimental setting, where all methods must operate under identical training data constraints.
>
> We agree that incorporating such methods—especially those that exploit limited OOD samples—could further enhance long-horizon stability. Exploring how GGND can benefit from these strategies without compromising fairness or data efficiency is an exciting direction that we plan to pursue in future work.
>
>
> # Response to Question 3
> > Model Limitations:
> > The theoretical analysis shows GGND's ability to manage geometric topological shifts. Are there any specific types of conformational shifts or molecular behaviors that GGND might still struggle with, even with its diffusion mechanism?
>
> We sincerely appreciate this insightful question. It indeed highlights an important broader topic—the characterization and detection of conformational shifts or OOD molecular behaviors.
>
> In our current work, GGND is designed to address geometric topological shifts arising from local structural distortions. However, it may still face challenges in modeling rare-event dynamics that require specialized long-timescale treatments (e.g., transition-state crossings or slow collective rearrangements). Curating relevant benchmarking and exploring how GGND can be extended to better capture these phenomena will be a focus of future work.

---

### Official Review · Reviewer_eXFQ · 2025-11-01

**Soundness:** 3
**Presentation:** 3
**Contribution:** 3
**Rating:** 6
**Confidence:** 2

**Summary:**

The paper proposes Geometric Graph Neural Diffusion (GGND), a module augments local SE(3)-equivariant GNNs with a global graph diffusion process to propagate information across all atom pairs while preserving equivariance.  GGND introduces equivariant gradient and equivariant diffusivity (attention) operators, aiming to learn features that are invariant to geometric–topological shifts to improve long-horizon MD stability beyond in-distribution accuracy. Experiments on 3BPA and SAMD23 report gains in energy/force MAE and higher stability in OOD scenarios.

**Strengths:**

*  GGND is a plug-in-play diffusion module that can integrate with common equivariant GNN backbones, preserves equivariance, and facilitates global information mixing.

* On 3BPA across 600–1200 K, GGND increases stable rollout length by orders of magnitude versus baselines while simultaneously reducing energy/force MAE. On SAMD23 (SiN/HfO), indicating robustness well beyond in-distribution regimes.

**Weaknesses:**

Disclaimer: The theory analysis of the paper is beyond my expertise, so I’m open to the authors’ clarifications and to other reviewers’ perspectives.

The theorem 3.1 justifies how geometric graph neural diffusion is able to reduce the representation variation of the graph neural network, but I'm generally curious why (and when) graph neural diffusion is superior over  equivariant attention models (e.g., Equiformer, TorchMD-Net) once those baselines are given sufficient effective depth/receptive field. A discussion or an ablation study would help clarify the conditions under which GGND is actually superior.

**Questions:**

What is the compute and memory increase when adding GGND to GNN backbone?

---

> ### Author Response · Authors · 2025-11-21
> **First Response to Reviewer eXFQ**
>
> # Response to Reviewer eXFQ
>
> We sincerely appreciate the time and effort of the reviewers in providing their valuable feedback. We are pleased that the reviewer recognized the strength of our paper and appreciated our experiments.
>
> # Response to Weakness 1
> > The theorem 3.1 justifies how geometric graph neural diffusion is able to reduce the representation variation of the graph neural network, but I'm generally curious why (and when) graph neural diffusion is superior over equivariant attention models (e.g., Equiformer, TorchMD-Net) once those baselines are given sufficient effective depth/receptive field. A discussion or an ablation study would help clarify the conditions under which GGND is actually superior.
>
> Thank you for this excellent question. While equivariant attention models can, in principle, achieve large effective receptive fields by increasing depth, in practice, they tend to suffer from over-squashing and over-smoothing as depth grows. Prior work on graph neural diffusion (Chamberlain et al., 2021) shows that PDE-inspired diffusion operators propagate information globally in a controlled, smooth manner, alleviating these issues and enabling substantially deeper architectures without degradation. Besides, our method employs the combination of local EGNN with our GGND, ensuring the characterization of local environment information while addressing the topological shifts.
>
> To further address the reviewer’s concern, we conducted additional experiments where Equiformer-V2 was given a larger cutoff radius and an increased maximum number of neighbors. Since the recommended Equiformer-V2 configuration already contains 20 layers, which is larger than ours, we did not increase the depth.
>
> The results (included in the revised submission) show that increasing the effective receptive field of Equiformer-V2 does not improve stability in OOD settings. In fact, the fully-connected Equiformer-V2 variant performs worse sometimes, which we attribute to its reduced ability to capture local environments—a benefit from GNN-based methods.
>
>
> | | | Metrics | Equiformer V2 | Equiformer V2 | GGND |
> | - | - | - | - | - | - |
> | | Parameters | Layer | 20 | 20 | 9 |
> | | | Cutoff | 12 | 20 | \- |
> | | | Maximum number of neighbors | 20 | 50 | \- |
> | SiN | Test | E/A | 0.010 | 0.012 | **0.009** |
> | | | F | 0.451 | 0.458 | **0.443** |
> | | | S | 98.284 | 97.931 | **100** |
> | | OOD | E/A | 0.021 | 0.035 | **0.015** |
> | | | F | 0.972 | 1.024 | **0.754** |
> | | | S | 82.031 | 75.834 | **99.892** |
> | HfO | Test | E/A | **0.005** | 0.007 | **0.005** |
> | | | F | 0.298 | 0.225 | **0.179** |
> | | | S | 97.184 | 99.278 | **100** |
> | | OOD | E/A | 0.010 | 0.011 | **0.008** |
> | | | F | 0.683 | 0.832 | **0.279** |
> | | | S | 79.762 | 68.235 | **97.928** |
>
> # Response to Questions:
> > What is the compute and memory increase when adding GGND to GNN backbone?
>
> We appreciate this important question. Below, we summarize the compute and memory overhead observed when adding GGND to ViSNet. All models are trained with identical hyperparameters; ViSNet* includes an additional 500 training epochs for a more comprehensive comparison. Experiments were conducted on an NVIDIA GeForce RTX 3090 GPU with an Intel(R) Xeon(R) Platinum 8338C CPU.
>
> Overall, GGND increases training time by 26.54% and MD simulation time by 15.57%. GPU memory usage increases by 15.28% during training and 14.87% during inference. Given the substantial improvements in energy/force accuracy and stability, we consider these additional computational and memory costs to be practical.
>
>
> | | Metrics | VisNet | VisNet* | +GGND |
> | - | - | - | - | - |
> | Epoch | | 1000 | 1500 | 1000 |
> | Time | Training Time (h) | 6.822 | 10.326 | 8.633 |
> | | Inference Time (s) | 13.872 | 13.672 | 16.032 |
> | | MD Time for 100 ps (h) | 1.358 | 1.349 | 1.582 |
> | Space | Training Memory (GiB) | 20.457 | 20.455 | 23.582 |
> | | Inference Memory (GiB) | 14.125 | 14.248 | 16.225 |
> | 1200K | Enery (eV/Å) | 3.464 | 3.193 | 0.583 |
> | | Forces (eV/A) | 1.404 | 1.383 | 0.304 |
> | | Stability (ps) | 0.004 | 0.009 | 11.209 |

---

### Official Review · Reviewer_sTeQ · 2025-11-08

**Soundness:** 2
**Presentation:** 3
**Contribution:** 2
**Rating:** 4
**Confidence:** 5

**Summary:**

The manuscript proposes Geometric Graph Neural Diffusion (GGND), a plug-in SE(3)-equivariant module that performs fully connected, diffusion-style updates on atom features and is then fused with a local equivariant GNN backbone. The central claim is that this global diffusion makes node representations insensitive to "geometric topological shifts" caused by temperature-dependent conformations and cutoff-dependent neighbor graphs, thereby improving out-of-distribution (OOD) stability in molecular dynamics. Formally, the model evolves features by a PDE-like update, $\partial_t Z(t)=(S(Z, X, t)-I) Z(t)$, and the paper presents a regret-style bound to argue reduced OOD error under adjacency shifts. Experiments on 3BPA (train at 300K, test at 600/1200K and dihedral slices) and on the SAMD23 SiN/HfO benchmark report notable gains in both force/energy MAE and a stability metric based on long-rollout NVE simulations up to 100 ps . The manuscript positions GGND as a general "plug-and-play" layer to retrofit existing local EGNNs.

**Strengths:**

From the angle of originality, the paper makes a concrete architectural move: it injects an equivariant, all‑pairs diffusion operator into otherwise local, cutoff‑dependent message passing, with spherical harmonics and Clebsch–Gordan structure to preserve symmetry. This is not only technically consistent with modern E(3)/SE(3) practice but is also aligned with a well‑motivated failure mode: changing neighbor graphs across temperatures or cutoffs degrade MLFFs even when in‑distribution errors look small. The formalization of “geometric topological shift” and the decomposition of the OOD gap give the work a clearer problem statement than usual in this area. The empirical section goes beyond pointwise MAE to include trajectory stability under NVE, which, in the opinion of the reviewer, is the right stress test to separate superficially accurate models from those that do not explode in practice.

**Weaknesses:**

The theoretical core reads fragile. In §3.1 the diffusivity $S(Z, X, t)$ is described as dependent on the adjacency $A$, which matches the paper's narrative about topology shift; however, the proof of Theorem 3.1 later asserts the model "assumes full connectivity" and that $S$ depends only on positions/features and thus is independent of $A$, hence $Z(T)$ does not depend on $A$ at all. Under that assumption, the main bound collapses to a trivial statement. This contradiction must be resolved unambiguously, because the claimed robustness hinges on the exact role of $A$.

The reporting has unit inconsistencies that undermine trust. Table 1 explicitly states energy MAE in eV, yet the surrounding text speaks of meV for the same numbers and conclusions (e.g., improvements claimed “below chemical accuracy”). This is a three‑orders‑of‑magnitude discrepancy that changes the physical interpretation and must be corrected before one can judge the effect size.

The experimental design does not yet establish that the new diffusion mechanism is the decisive factor. The ablation includes a “fully‑connected message passing” variant, but there is no head‑to‑head against strong global equivariant baselines that already propagate information across all pairs, such as Allegro (strictly local but designed to capture many‑body correlations without message passing) and modern equivariant Transformers like EquiformerV2. Without those comparisons, one cannot attribute the OOD gains specifically to the PDE‑style update rather than to the mere presence of global interactions. A fair suite here would at least include NequIP and MACE (already present) plus Allegro and EquiformerV2, all tuned for the same cutoff and time step.

The evaluation scope feels narrow for the strength of the claims. Stability is reported up to 100 ps in NVE with Velocity‑Verlet and a 1 fs step; that is useful but short. The paper should probe longer horizons, and also include NVT (Nosé‑Hoover and Langevin) to show the method is not sensitive to ensemble or integrator/thermostat coupling. The “Forces are not Enough” paper makes exactly this point about trajectory‑level validation; aligning with that protocol would improve credibility.

Related work is incomplete in a way that directly affects positioning. GG‑ODE (KDD’23) explicitly addresses cross‑environment dynamics by learning a shared graph‑neural ODE with environment‑specific latent exogenous factors (e.g., temperature), a complementary strategy to the paper’s “invariance to topology change.” It is not discussed or compared, and the reader cannot see whether conditioning on environment would reduce or even remove the need for diffusion on a fully connected graph.

Finally, scalability is acknowledged as $O\left(N^2\right)$ but not quantified. If the core is all-pairs diffusion, memory and wall-clock scaling should be reported on realistic MD supercells, not only 27-atom 3BPA or 96-atom HfO cells. A convincing story would include sparsification or hierarchical diffusion and a clear regime where GGND remains practical.

**Questions:**

I would like to see precise clarifications to the following questions. First, is $S$ ever computed from a sparse $A$ in training or inference, or is the operator strictly complete-graph at all times? Please fix the contradiction between §3.1 and the proof of Theorem 3.1, and state the exact dependence on $A$ in the method section. Second, please correct units in every table and paragraph and re-evaluate claims like "below chemical accuracy"; the present mix of $\mathrm{eV} / \mathrm{meV}$ is not acceptable. Third, what happens beyond 100 ps and under thermostats? Showing stability curves up to $\geq 1 \mathrm{~ns}$ and including NVT/NPT would reduce concerns about integrator-specific artefacts. Fourth, why is there no comparison to Allegro and EquiformerV2 as strong global baselines? A single paragraph in Related Work is not sufficient; the comparison needs to be empirical on the same splits, with matched cutoffs, time steps, and neighbor policies. Fifth, can the authors release scripts for the stability metric (bond- and RDF-based) and the exact MD settings (neighbor list rebuilds, cutoffs, barostat/thermostat where applicable), since small choices there can create large apparent gaps? Finally, is there a way to incorporate environment conditioning, something like GG-ODE, on top of GGND to see whether invariance and conditioning are complementary rather than alternatives?

---

> ### Author Response · Authors · 2025-11-21
> **First Response to Reviewer sTeQ [1/2]**
>
> # Response to Reviewer sTeQ
>
> We thank the reviewer for the thoughtful and detailed assessment. We especially appreciate the recognition of the paper’s originality, the formalization of geometric topological shifts, and the emphasis on trajectory-level validation. Below, we address each concern and provide precise clarifications, corrections, and additional results.
>
> # Response to Weakness 1 and Question 1
> > First, is  ever computed from a sparse  in training or inference, or is the operator strictly complete-graph at all times? Please fix the contradiction between §3.1 and the proof of Theorem 3.1, and state the exact dependence on  in the method section.
>
> We sincerely apologize for the confusion caused by the mixing of notations for the adjacent matrix. Our methods involve two adjacent matrices: 1) a local connected graph determined by the environment and the model ($\mathbf{A}$), and 2) a fully-connected graph ($\mathbf{A}_g$), which are not clearly highlighted before.
>
> We would like to clarify that:
> 1. The GGND diffusion operator $\mathbf{S}(\mathbf{Z}(t), \mathbf{X}, t)$ is always applied on a complete graph ($\mathbf{A}_g$);
> 2. $\mathbf{Z}(t)$ learned by GGND depends on $\mathbf{A}_g$ and does not depend on $\mathbf{A}$.
>
> We have revised the submission accordingly.
>
> # Response to Weakness 2 and Question 2
> > Second, please correct units in every table and paragraph and re-evaluate claims like "below chemical accuracy"; the present mix of  is not acceptable.
>
> We agree that the mixed use of eV and meV is unacceptable and apologize for the oversight.
> Corrections to be made:
> - All energy MAEs will be uniformly reported in eV;
> - All force MAEs will be corrected in eV/Å;
> - Claims about “below chemical accuracy” will be re-evaluated and corrected.
>
> We have updated all tables, captions, and text to ensure full consistency.
>
>
> # Response to Weakness 4 and Question 3
> > Third, what happens beyond 100 ps and under thermostats? Showing stability curves up to and including NVT/NPT would reduce concerns about integrator-specific artefacts.
>
>
> We appreciate the reviewer’s insightful question. To mitigate concerns about integrator-specific artifacts, we have included *classical MD* as an additional baseline in our stability curves. We further extended our analysis to **200 ps** and added simulations under the **NVT ensemble**, with the full results provided in the revised submission.
>
> Our findings show that the behavior of classical MD under thermostats is consistent with that of our setup. Moreover, GGND maintains substantially improved stability compared to all baselines across both **NVE** and **NVT** molecular dynamics simulations.
>
>
> # Response to Weakness 3 and Question 4
> > Fourth, why is there no comparison to Allegro and EquiformerV2 as strong global baselines? A single paragraph in Related Work is not sufficient; the comparison needs to be empirical on the same splits, with matched cutoffs, time steps, and neighbor policies.
>
> We appreciate the reviewer’s perspective. We would like to clarify that neither Allegro nor EquiformerV2 performs message passing on a fully connected graph. Besides, neither of these two methods addressed stability or generalization. Thus, we did not include these two baselines before. Here, we added these two baselines
>
> | Molecule | Splits | Metrics | Allegro | Equiformer V2 | GGND |
> | - | - | - | - | - | - |
> | SiN | Test | E/A | 0.015 | 0.010 | **0.009** |
> | | | F | 0.673 | 0.451 | **0.443** |
> | | | S | 63.583 | 98.284 | **100** |
> | | OOD | E/A | 0.028 | 0.021 | **0.015** |
> | | | F | 1.185 | 0.972 | **0.754** |
> | | | S | 55.824 | 82.031 | **99.892** |
> | HfO | Test | E/A | 0.007 | **0.005** | **0.005** |
> | | | F | 0.385 | 0.298 | **0.179** |
> | | | S | 64.282 | 97.184 | **100** |
> | | OOD | E/A | 0.012 | 0.010 | **0.008** |
> | | | F | 0.593 | 0.683 | **0.279** |
> | | | S | 60.982 | 79.762 | **97.928** |
>
> The table demonstrates that EquiformerV2 exhibits competitive performance on the in-distribution testing set; however, its performance on the OOD set remains unsatisfactory. These baselines have been included in our revised submission.

---

> ### Author Response · Authors · 2025-11-21
> **First Response to Reviewer sTeQ [2/2]**
>
> # Response to Question 5
> > Fifth, can the authors release scripts for the stability metric (bond- and RDF-based) and the exact MD settings (neighbor list rebuilds, cutoffs, barostat/thermostat where applicable), since small choices there can create large apparent gaps?
>
> We agree that replicability is essential, especially for MD trajectory analysis.
> We commit to publicly releasing:
> - scripts for calculating the stability metric;
> - scripts for running MD simulation;
>
> We have included these scripts in the ```evaluation``` directory in the supplementary repository.
>
> # Response to Weakness 5 and Question 6
> > Finally, is there a way to incorporate environment conditioning, something like GG-ODE, on top of GGND to see whether invariance and conditioning are complementary rather than alternatives?
>
> We appreciate this insightful suggestion. We have explored adding environment conditioning to GGND. However, we found that energy distributions across different temperatures are extremely sparse, making it challenging to train conditioning-based MLFF models reliably in our setting. This difficulty extends to approaches inspired by GG-ODE, which conditions on environment variables but evaluates on trajectories rather than energy/force.
>
> Nevertheless, we agree that invariance (GGND) and conditioning (GG-ODE-style) are conceptually complementary. We will discuss this paper in our revision. We remain enthusiastic about this direction and plan to investigate hybrid approaches in future work.
>
> # Response to Weakness 6
> > Finally, scalability is acknowledged as $O\left(N^2\right)$ but not quantified. If the core is all-pairs diffusion, memory and wall-clock scaling should be reported on realistic MD supercells, not only 27-atom 3BPA or 96-atom HfO cells. A convincing story would include sparsification or hierarchical diffusion and a clear regime where GGND remains practical.
>
> We appreciate this important comment. We would like to highlight that the experiments on SAMD23 contain molecules with atoms up to 510, which is comparatively large-scale in the literature. Additionally, there are no other available benchmarks that contain molecular data from different environments. Lastly, we appreciate the sparsification or hierarchical diffusion as a promising way to improve the scalability; however, we admit that achieving stability and scalability in one paper is challenging, and we would like to defer this as our future work.
>
> To alleviate the reviewer's concern, we summarize the compute and memory overhead observed when adding GGND to ViSNet. All models are trained with identical hyperparameters; ViSNet* includes an additional 500 training epochs for a more comprehensive comparison. Experiments were conducted on an NVIDIA GeForce RTX 3090 GPU with an Intel(R) Xeon(R) Platinum 8338C CPU.
>
> Overall, GGND increases training time by 26.54% and MD simulation time by 15.57%. GPU memory usage increases by 15.28% during training and 14.87% during inference. Given the substantial improvements in energy/force accuracy and stability, we consider these additional computational and memory costs to be practical.
>
> | | Metrics | VisNet | VisNet* | +GGND |
> | - | - | - | - | - |
> | Epoch | | 1000 | 1500 | 1000 |
> | Time | Training Time (h) | 6.822 | 10.326 | 8.633 |
> | | Inference Time (s) | 13.872 | 13.672 | 16.032 |
> | | MD Time for 100 ps (h) | 1.358 | 1.349 | 1.582 |
> | Space | Training Memory (GiB) | 20.457 | 20.455 | 23.582 |
> | | Inference Memory (GiB) | 14.125 | 14.248 | 16.225 |
> | 1200K | Enery (eV/Å) | 3.464 | 3.193 | 0.583 |
> | | Forces (eV/A) | 1.404 | 1.383 | 0.304 |
> | | Stability (ps) | 0.004 | 0.009 | 11.209 |

---

### Author Response · Authors · 2025-12-02
**Summary of Discussions**

Dear Program Chairs, Senior Area Chairs, and Area Chairs,

We sincerely appreciate the committee’s time and effort in reviewing our work.

Our paper proposed Geometric Graph Neural Diffusion to improve extrapolation to unseen conformations and enhance stability in molecular dynamics simulations. We received reviews from five reviewers, with scores of 4 (sTeQ), 6 (eXFQ), 8 (WP7J), 8 (Xkok), and 2 (Ev5w). We thank Reviewer sTeQ for recognizing our `concrete architectural contribution, clear problem formulation, and right evaluation metrics`. We appreciate Reviewer eXFQ for highlighting `the plug-and-play nature of our method and the robustness of our experiments`. We thank Reviewer WP7J for noting the `novelty, effectiveness, and practical value of our approach`, and Reviewer Xkok for emphasizing `the significance of the problem, the novelty of the method, and the overall quality and clarity of the paper`. We also thank Reviewer Ev5w for acknowledging `the solid theoretical foundations and useful experimental results`.

Below, we summarize the key points of our rebuttal:
### Expanded Discussion and Clarifications
* Scalability
* Model details
* Scope of generalization
* Notation and unit inconsistencies
### Additional Experiments
* Computational overhead
* Extended stability analysis
* New baselines

A more detailed response summary is provided below:

---
## 1. Expanded Discussions
### 1.1 Scalability
We explicitly acknowledge the O(N²) diffusion cost as a limitation. We have verified our method on 510-atom systems; we believe this scale is comparatively large compared to MLFF standards. We also discussed the extension to a larger scale.
### 1.2 Model Details
We added a detailed explanation of why PDE-style diffusion mitigates oversquashing/oversmoothing compared to fully connected MPNNs and clarified the relationship between diffusion operators and geometric topology.
### 1.3 Conditioning Method
We expanded the discussion on the complementarity between our method and environment-conditioning approaches (e.g., GG-ODE), and clarified why conditioning-based MLFFs are difficult to train due to sparse energy distributions across temperatures.
### 1.4 Generalization
We clarified that the scope of this work was generalization under geometric topological shifts within a single molecular system, rather than across different molecular systems.
## 2. New Experiments and Baselines
### 2.1 Computational Overhead
We added detailed wall-clock and memory comparisons for training, inference, and MD simulation. GGND incurs 26.54% training overhead, 15.57% MD overhead, and ~15% additional GPU memory. These costs are modest relative to the substantial gains—**83.16%** improvement in energy accuracy and a **2,800×** increase in stability.
### 2.2 Extended Stability Analysis
We extended NVE stability curves to 200 ps, added NVT simulations, and incorporated classical MD baselines to isolate integrator effects. GGND remains significantly more stable compared to baselines.
### 2.3 Strong Global Baselines
We added Allegro and EquiformerV2. Our proposed GGND matches or outperforms both in-distribution and exhibits substantial gains in OOD stability.
### 2.4 Enlarged-Receptive-Field EquiformerV2
We increased EquiformerV2’s cutoff radius and maximum neighbors. These changes do not improve stability and sometimes degrade accuracy, indicating that simply enlarging receptive fields cannot address instability induced by topological shift.

## 3. Clarifications and Corrections
* Resolved confusion between the local adjacency matrix $\mathbf{A}$ and the complete one $\mathbf{A}_g$.
* Standardized all units and corrected related claims.
## 4. Reproducibility
We released all scripts used for stability metrics and MD configurations in the supplementary repository.
## 5. Additional Visualizations
We added high-temperature conformation visualizations as requested.
## 6. Reviewer Highlights
* For Reviewer `sTeQ`, we addressed all concerns by clarifying the adjacency and complete-graph assumptions in Theorem 3.1, correcting units, extending stability analyses to 200 ps under NVE/NVT with classical MD baselines, and adding strong global baselines to demonstrate that our GGND’s OOD gains arise from PDE-style diffusion. We also released all stability and MD scripts and discussed complementarity with environment-conditioning approaches.
* For Reviewer `Ev5w`, we addressed all points by clarifying GGND’s link to geometric topology, correcting interpretability concerns, extending stability analyses to 200 ps under NVE/NVT with classical MD baselines, and quantifying computational overhead, showing modest cost relative to the stability and accuracy gains. We also acknowledged scalability limits, added high-temperature conformation visualizations, and discussed the notions of generalization.

We hope the addressed concerns and the additional evidence help the AC/SAC/PC evaluate the submission favorably.

Best regards,

Authors of Submission 3313

---

### Meta-Review · Area_Chair_4tj1 · 2026-01-07

**Summary:**

This paper introduces Geometric Graph Neural Diffusion (GGND), a plug-and-play module designed to enhance the stability of Molecular Dynamics (MD) simulations driven by Graph Neural Networks. The core innovation is augmenting local equivariant message passing with a global (fully connected) equivariant diffusion process. This allows the model to capture long-range correlations and "geometric topological shifts" (conformational changes at high temperatures) that local models typically fail to generalize to.
The initial reviews were divided, with two enthusiastic accepts (WP7J, Xkok), one weak accept with low confidence (eXFQ), one negative review questioning he validity of the problem setting (Ev5w), and one weak reject (sTeQ) that raises technical critiques.

Most of the reviewers appreciated the novelty of the architectural design, the theoretical formulation of the "topological shift" problem, and the empirical gains in simulation stability on the OOD test set from 3BPA and SAMD23 benchmarks.  Major concerns concentrated on (1) scalability issue (all reviewers) where the fully-connected graph result in $O(N^2)$ complexity, (2) missing baselines (sTeQ, eXFQ) such as EquiformerV2 and Allegro, and (3) scope of generalization being limited to specific type of distortion (WP7j, Ev5w), and (4)imperfect experiment setting: insufficient simulation time and integrator-specific artefacts (sTeQ), choice of stability metric (sTeQ, Ev5w), non-standard MD settings (Ev5w).  sTeQ raised concerns about method clarity and inconsistent units used in reported results.

**Reviewer Concerns:**

Overall the rebuttal was effective in addressing technical concerns:

**Addressed concerns**:
- More baselines(sTeQ, eXFQ): The authors added comparisons to Allegro and EquiformerV2, and provide justification of the diffusion approach against other global models without diffusion.
- Method clarity and unifying unit (sTeQ): The contradiction regarding the adjacency matrix $A$ vs. $\bar{A}$ was resolved, and the severe unit inconsistencies (eV vs. meV) were corrected.
- Interpretability (Ev5w): The authors added visualizations of the structures being generated.

**Partially addressed concerns**:
- Validity of stability evaluation (sTeQ, eXFQ): The authors extended simulations to 200ps, included NVT ensembles, and added Classical MD baselines to rule out integrator artifacts. But sTeQ suggested >1ns simulations, and eXFQ requested more metrics for stability; these are not directly satisfied in the rebuttal.
- Scalability(all reviewers): The authors provided a detailed breakdown showing a ~26% training overhead and ~16% inference overhead on the benchmark datasets. However, they also acknowledge the $O(N^2)$ complexity limits application to medium-sized systems (e.g., <1000 atoms). This is an accepted limitation for the current contribution.

**Outstanding concerns**:
- Limited generalization mode(WP7j, Ev5w): the authors admitted that "GGND is designed to address geometric topological shifts arising from local structural distortions" and may still face challenges in modeling other types of conformational changes.

**Reviewer Scores:**

Based on the rebuttal, I'd predict that reviewers with initial positive scores would maintain their rating, while reviewer sTeQ may increase their score from 4 to 6, and reviewer Ev5w remains slightly negative.

- Reviewer sTeQ (Original: 4): Predicted: 6. The rebuttal addressed their concerns on method clarity, inconsistent unit, added requested baselines (Allegro and EquiformerV2), and partially addressed the evaluation setting (increase to 200ps).
- Reviewer eXFQ (Original: 6): Predicted: 6. The initial review was low-confidence and mainly requested an explanation on why diffusion will provide benefit.
- Reviewer WP7J (Original: 8): Predicted: 8 (Accept). Remained positive; concerns about scalability were noted as future work.
- Reviewer Xkok (Original: 8): Predicted: 8 (Accept). Remained positive. The original review was short and already very positive.
- Reviewer Ev5w (Original: 2): Predicted: 2 or 4. Unlikely to switch sides as this reviewer fundamentally disagrees with the experimental setup (high-temperature stress tests) and the scope of claimed "OOD generalization".

---

> ### Public Comment · ~Haokai_Hong1 · 2026-02-28
> **Request for Minor Title Update for Camera-Ready Submission**
>
> Dear Area Chair and Reviewers,
>
> We sincerely appreciate your time in reviewing our paper.
>
> To better highlight our work, we would like to request a minor title update for the camera-ready version: from __Geometric Graph Neural Diffusion for Stable Molecular Dynamics__ to __Geometric Graph Neural Diffusion for Stable Molecular Dynamics ```Simulations```__. We believe this wording change does not significantly change the scope of our paper.
>
> Please let us know if this update is acceptable or if any further steps are required.
>
> Thank you for your time and consideration.
>
> Best regards,
>
> Authors

---

### Decision · Program_Chairs · 2026-01-26

Accept (Poster)